# Strength of interactions in the Notch gene regulatory network determines patterning and fate in the notochord

Héctor Sánchez-Iranzo[1]*[†], Aliaksandr Halavatyi[2], Alba Diz-Muñoz[1]*

[1]Cell Biology and Biophysics Unit, European Molecular Biology Laboratory, Heidelberg, Germany; [2]Advanced Light Microscopy Facility, European Molecular Biology Laboratory, Heidelberg, Germany

**Abstract** Development of multicellular organisms requires the generation of gene expression patterns that determines cell fate and organ shape. Groups of genetic interactions known as Gene Regulatory Networks (GRNs) play a key role in the generation of such patterns. However, how the topology and parameters of GRNs determine patterning *in vivo* remains unclear due to the complexity of most experimental systems. To address this, we use the zebrafish notochord, an organ where coin-shaped precursor cells are initially arranged in a simple unidimensional geometry. These cells then differentiate into vacuolated and sheath cells. Using newly developed transgenic tools together with *in vivo* imaging, we identify *jag1a* and *her6/her9* as the main components of a Notch GRN that generates a lateral inhibition pattern and determines cell fate. Making use of this experimental system and mathematical modeling we show that lateral inhibition patterning is promoted when ligand-receptor interactions are stronger within the same cell than in neighboring cells. Altogether, we establish the zebrafish notochord as an experimental system to study pattern generation, and identify and characterize how the properties of GRNs determine self-organization of gene patterning and cell fate.

**\*For correspondence:**
hector.sanchez@kit.edu (HS-I);
diz@embl.de (AD-M)

**Present address:** [†]Institute of Biological and Chemical Systems – Biological Information Processing, Karlsruhe Institute of Technology, Eggenstein-Leopoldshafen, Germany

**Competing interest:** The authors declare that no competing interests exist.

## Editor's evaluation

This manuscript presents computational and experimental results to study lateral inhibition patterning in the zebrafish notochord, identifying Jag1a as a crucial ligand and marker for vacuolated cell fate whereas her6 and her9 repress Jag1a. The results are complemented with numerical simulations of lateral induction and lateral inhibition circuits in one-dimensional arrays, together with linear stability analysis. The work is very well done and makes an important contribution to the understanding of notochord development.

## Introduction

Most of the information necessary to build an organism resides in its genome. The co-regulation of subsets of genes form gene regulatory networks (GRNs) that generate patterns of expression, which ultimately regulate cell fate and organ shape. Different types of GRNs regulate different patterning events. For example, some GRNs work in combination with gradients of morphogens to generate patterns at the embryo or organ scale (*Briscoe and Small, 2015*). In contrast, other GRNs coordinate short-range interactions, generating self-organized patterns of gene expression at the cellular scale (*Schweisguth and Corson, 2019*; *Sjöqvist and Andersson, 2019*). Understanding how different GRN topologies and the strength of their interactions regulate the generation of gene expression patterns

constitutes a key challenge in developmental biology. However, research in this direction has been hindered by limited experimental systems that can be accurately modeled mathematically.

GRNs controlling short-range interactions produce diverse patterning events, such as lateral inhibition and lateral induction. Lateral inhibition involves a group of cells actively suppressing the expression of some genes in adjacent cells, thereby inducing them to adopt a different cell fate. In contrast, lateral induction involves cells inducing adjacent cells to adopt the same cell fate. Lateral inhibition and lateral induction patterns are two of the main patterns generated by Notch GRNs: one of the most representative signaling pathways that mediates local communication between cells. The Notch pathway is evolutionarily conserved and generates gene expression patterns that regulate cell fate decisions in a wide variety of organs (*Apelqvist et al., 1999*; *Artavanis-Tsakonas et al., 1999*; *VanDussen et al., 2012*; *Wibowo et al., 2011*). Signaling is triggered by interaction of a Notch receptor with a Notch ligand. Once they bind, the Notch intracellular domain (NICD) is cleaved inside the signal receiving cell, and in complex with Rbp-Jκ and MAML, translocates to the nucleus, where it activates Notch target genes (*Bray, 2016*).

The generation of either lateral inhibition or lateral induction patterns downstream of Notch has thus far been associated with different ligands. Lateral inhibition patterning has been described for the Delta-like (Dll) ligands and for *Jag2* (*Heitzler and Simpson, 1991*; *Lanford et al., 1999*) and generally occurs when Notch signaling activates the expression of a transcriptional repressor of the HES family that in turn inhibits the expression of the ligand in adjacent cells, preventing them from adopting the same cell fate (*Simpson, 1990*; *Sjöqvist and Andersson, 2019*; *Sternberg, 1988*). Mathematical simulations have shown that a lateral inhibition GRN can amplify small levels of noise in gene expression, leading to bi-stability and the generation of alternating patterns (*Collier et al., 1996*). Lateral induction has been shown for the ligand *Jag1*, whereby Notch activation by *Jag1* triggers the expression of the same ligand in the adjacent cells, promoting the same fate (*Hartman et al., 2010*; *Manderfield et al., 2012*; *Neves et al., 2011*). It remains unknown whether lateral inhibition and lateral induction GRNs are restricted to specific ligands, or whether a given ligand can generate different patterns depending on the cellular and signaling context.

Other important parameters in a GRN are the nature and affinities of the ligand-receptor interactions. In the case of Notch, ligands can also interact with receptors in the same cell (*Celis de and Bray, 1997*; *Klein et al., 1997*; *Micchelli et al., 1997*). This interaction, known as cis-inhibition, mutually inactivates both the ligand and receptor, and mathematical models have shown that it is required for patterning in the absence of cooperative interactions (*Formosa-Jordan and Ibañes, 2014*; *Sprinzak et al., 2010*; *Sprinzak et al., 2011*). Different ligands and receptors bind to each other in cis and trans with different affinities, and these affinities can be modulated by posttranslational modifications (*Bray, 2016*; *Sjöqvist and Andersson, 2019*). Altogether, these properties increase the complexity and diversity of Notch GRNs. For this reason, understanding how the topology and interaction parameters of these GRNs lead to pattern generation requires a combination of mathematical models and experimental systems that allow *in vivo* visualization and perturbation of Notch signaling components.

The notochord constitutes an underappreciated system that is ideal for studying the generation of Notch patterns. Initially, notochord coin-shaped precursor cells are arranged unidimensionally. These simple and well-defined cell-cell contacts greatly facilitate mathematical modeling and theoretical analysis, making it valuable for studying the relationship between GRNs parameters and patterns. In vertebrates, such as zebrafish, notochord precursors give rise to two different cell types (*Dale and Topczewski, 2011*): vacuolated cells, located in the inner part of the organ, that contain a large vacuole that provides hydrostatic pressure (*Adams et al., 1990*; *Bagwell et al., 2020*; *Ellis et al., 2013*), and sheath cells, which form the surface of the cylindrical structure (*Dale and Topczewski, 2011*; *Grotmol et al., 2003*; *Figure 1A*). The cell fate decision between vacuolated and sheath cells depends on Notch signaling (*Yamamoto et al., 2010*). Inhibition of the Notch ligands *jag1a* and *jag1b* by morpholino (MO) injection leads to an excess of vacuolated cells, while overexpression of NICD promotes sheath cell fate (*Yamamoto et al., 2010*). However, most of the components and topology of the GRN that coordinates cell fate in the notochord remain unknown.

Here, we exploit the *in vivo* imaging and genetic manipulations that the zebrafish model offers to quantitatively study the generation of Notch patterns. We establish the zebrafish notochord as the first unidimensional system to study lateral inhibition patterning. Using this experimental model, we show that *jag1a* generates a lateral inhibition pattern, a possibility thought to be restricted to the

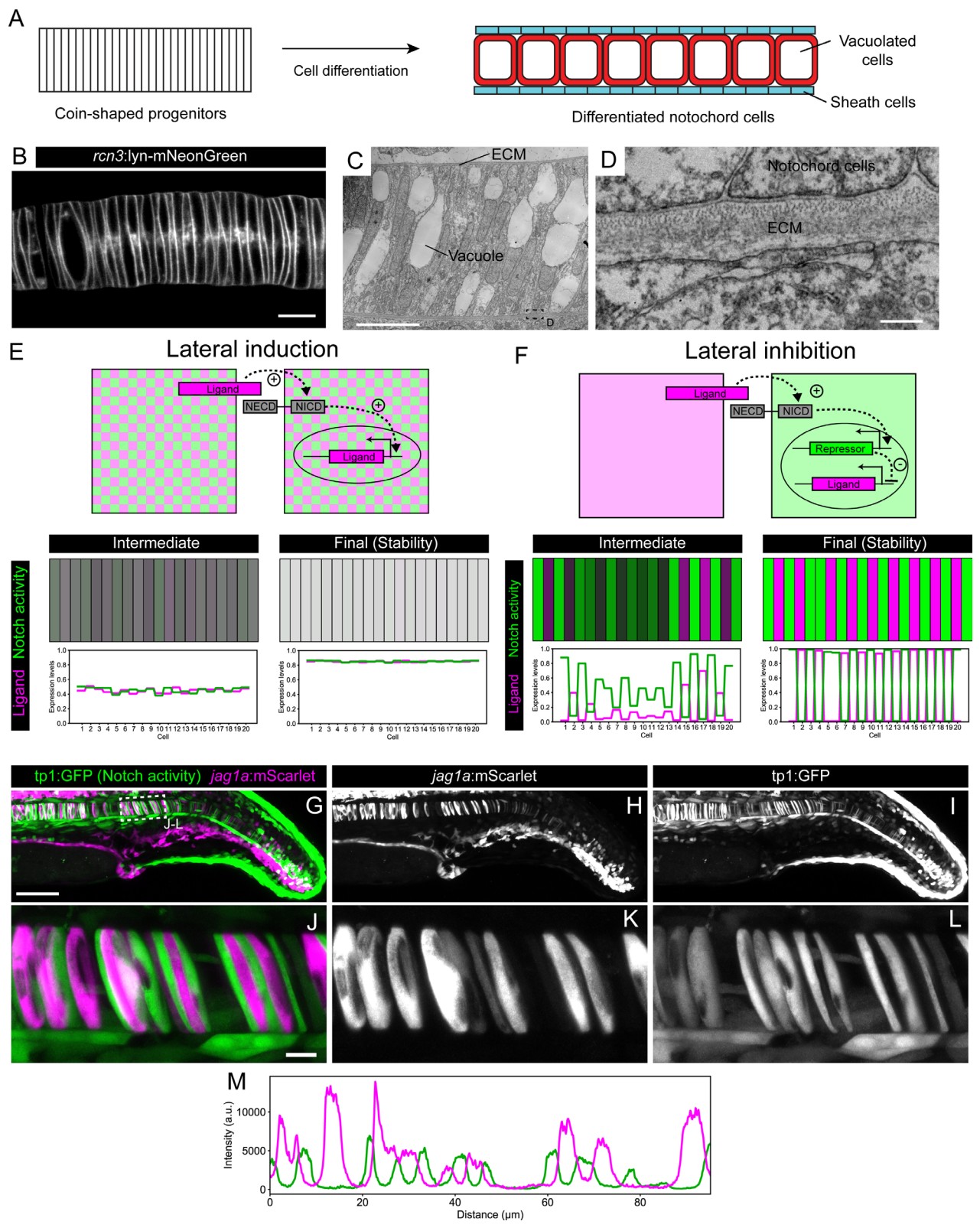

**Figure 1.** *Jag1a* generates a lateral inhibition pattern. (**A**) Schematic representation of notochord development. At 18–19 hpf most of the notochord is composed of coin-shaped precursor cells. During the following 8 hours, progressively, in an antero-posterior order, coin-shaped precursor cells begin their differentiation into sheath cells and vacuolated cells. (**B**) Airyscan confocal section of a zebrafish notochord at 19 hpf using the *rcn3*:lyn-mNeonGreen transgenic line. (**C**) Transmission electron microscopy of a zebrafish notochord at 19 hpf. (**D**) Magnification of boxed area in (**C**). (**E**) (Top)

*Figure 1 continued on next page*

*Figure 1 continued*

Schematic representation of the model for a Lateral Induction Network shows a pair of cells where the ligand in one cell activates NICD release in the neighboring cell. NICD activates ligand expression in its own cell. (Bottom) Representative simulation of this network applied to an array of cells unidimensionally arranged. (F) (Top) Schematic representation of the model for a Lateral Inhibition Network shows a pair of cells where the ligand in one cell activates NICD release in the neighboring cell. NICD activates the expression of the repressor, which in turn inhibits ligand expression. (Bottom) Simulation of this network applied to an array of cells unidimensionally arranged. (G–L) Maximal intensity projection of Airyscan confocal sections of a zebrafish tail at 22 hpf. (J–L) Magnification of boxed area in (G). n = 10 fish. (L) Intensity profile across a horizontal line in panel (L). (M) *jag1a*:mScarlet and tp1:GFP expression levels across a 1 μm thick horizontal line on a single plane of the image shown in J. Scale bars, 1 μm (D) 10 μm (B, C, J), 100 μm (G).

The online version of this article includes the following figure supplement(s) for figure 1:

**Figure supplement 1.** Lateral induction and lateral inhibition simulations, and mRNA expression pattern of *jag1a* and *jag1b*.

other Notch ligands (*Boareto, 2020*; *Boareto et al., 2015*; *Sjöqvist and Andersson, 2019*). Using a combination of single-cell RNA-Seq analysis and genetic perturbations, we identify *her6/her9* and *jag1a* as the key genes that promote sheath and vacuolated fate. Our computational modeling further reveals that a stronger cis- than trans-inhibition promotes the generation of lateral inhibition patterns. We experimentally validate the role of cis-inhibition in our GRN, finding that *jag1a* is sufficient to disrupt the expression of Notch-target genes in the cells where it is expressed. Altogether, our results describe and characterize a novel Notch GRN that generates lateral inhibition patterns and determines cell fate.

## Results

### *Jag1a* and Notch activity show a lateral inhibition pattern in the zebrafish notochord

Notch signaling generates patterns of gene expression by signaling at cell-cell contacts (*Bray, 2006*; *Shaya et al., 2017*). Thus, a prerequisite for the study of Notch patterning in the notochord is the characterization of cell-cell contacts. To describe the contacts between cells, we generated an *rcn3*:lyn-mNeonGreen transgenic line that labels the plasma membrane of all notochord cells. We observed that notochord precursor cells are coin-shaped and unidimensionally arranged one cell after another (*Figure 1B*). Using transmission electron microscopy, we confirmed this cell arrangement and observed that coin-shaped notochord cells are isolated from the rest of the tissues by a layer of extracellular matrix (*Figure 1C–D*). Thus, the contacts of each notochord cell are restricted to the two neighboring cells in the stack. This unidimensional geometry with very well-defined cell-cell contacts makes the notochord an ideal system to study Notch patterning.

Whether Notch signaling generates gene expression patterns in the notochord remains unknown. To understand the expression patterns that may be generated in this organ, we modeled lateral induction and lateral inhibition networks in the unidimensional arrangement of notochord cells. We first modeled a lateral induction network as a two component GRN, where the Notch ligand induces NICD cleavage in the adjacent cells, and NICD in turn induces ligand expression in the cells where it is located. This network gives rise to a homogeneous pattern, where all the cells have both high concentrations of NICD and ligand (*Figure 1E* and *Figure 1—figure supplement 1A*, *Matsuda et al., 2012*; *Petrovic et al., 2014*). Next, we modeled a lateral inhibition network (*Collier et al., 1996*). Here, the ligand also induces NICD cleavage in the adjacent cells, but in this case, NICD induces the expression of a repressor that in turn inhibits ligand expression. The result of this model is a NICD-ligand alternating pattern (*Figure 1F* and *Figure 1—figure supplement 1B*). These results are in agreement with previous models of lateral induction and lateral inhibition (*Collier et al., 1996*; *Matsuda et al., 2012*; *Petrovic et al., 2014*).

Then, we experimentally evaluated whether one of these two patterns was present in the notochord. The two zebrafish homologs of the mammalian *Jag1* – *jag1a* and *jag1b* – are the main Notch ligands in the notochord (*Yamamoto et al., 2010*). Although both jag1 ligands show a non-homogeneous expression pattern, the *jag1a* one is sharper and can be observed in more immature cells – more posteriorly in the notochord – (*Figure 1—figure supplement 1C-F*). For this reason, and to explore Notch patterns in high resolution, we generated a stable *jag1a*:mScarlet BAC transgenic line that recapitulates the endogenous *jag1a* mRNA expression (*Figure 1—figure supplement 1C-E*), and

crossed it to the tp1:GFP transgenic line (*Parsons et al., 2009*). The tp1 promoter includes 12 Rbp-Jκ binding sites derived from a viral sequence, making the tp1:GFP line a reporter of Notch activity. Interestingly, we found an alternating pattern (*Figure 1G–M*, *Figure 1—figure supplement 1G, H*) that resembles lateral inhibition, a pattern that has never been described for *Jag1*.

To verify that the observed pattern is generated by lateral inhibition, we injected previously validated (*Yamamoto et al., 2010*) splicing-*jag1a* and atg-*jag1b* MOs into the tp1:GFP;*jag1a*:mScarlet double transgenic line. By using a splicing-*jag1a* MO we specifically inhibited endogenous *jag1a* genes but not our *jag1a*:mScarlet reporter. First, we observed that when we injected the two MOs simultaneously, the tp1:GFP signal almost completely disappeared in the notochord, but not in the neighboring tissues (*Figure 2D*), supporting the hypothesis that *jag1a* and *jag1b* are the main, if not the only, Notch ligands expressed in the notochord. We also observed an increase in the number of *jag1a*:mScarlet-positive cells that are directly adjacent to other *jag1a*:mScarlet-positive cells, suggesting that a lateral inhibition process is disrupted upon inhibition of *jag1a* and *jag1b*. This effect was also observed, although to a lower extent, when injecting the *jag1a* or *jag1b* MOs separately, indicating that *jag1a* and *jag1b* have similar, but not completely redundant roles in the generation of the lateral inhibition pattern (*Figure 2A–E*).

Together, our results show that *Jag1* is not restricted to the generation of lateral induction patterns as previously thought, but can also generate lateral inhibition patterns.

### *Jag1a* and Notch activity are early markers of notochord cell fate

Finding early markers of differentiation is important to understand cell fate decisions. However, no early marker of notochord cell differentiation has been reported to date. Having identified an alternating tp1-*jag1a* pattern, we evaluated whether it is associated with vacuolated and sheath cell fates. To test this, we used the tp1:GFP;*jag1a*:mScarlet double transgenic reporter, and followed notochord cells by time lapse *in vivo* imaging (*Figure 2G* and *Figure 2—video 1*). We found that *jag1a*-positive cells gave rise to vacuolated cells, while tp1-positive cells differentiated into sheath cells (*Figure 2F*). Interestingly, at the end of the movie, most of the vacuolated cells are labeled with *jag1a*:mScarlet, while there are some non-labeled cells at the notochord surface. This suggests that the non-labeled cells at the disc-shape stage are Notch active and will differentiate into sheath cells, but their Notch activity is not strong enough to activate the non-endogenous tp1 promoter.

Having identified *jag1a* is an early marker for cell fate, we decided to verify if cell fate is determined by a lateral inhibition process. An important characteristic of lateral inhibition is that the cell expressing the ligand prevents the neighboring cells to acquire the same cell fate. To evaluate if this is the case in the notochord, we quantified how often two consecutive coin-shaped cells acquire vacuolated cell fate. To do this, we developed a feedback microscopy pipeline that allowed us to image the notochord cells in high quality over time, even though the fish was simultaneously elongating (*Figure 2—figure supplement 1* and *Figure 2—videos 2; 3*). We found that none of the future vacuolated cells were adjacent to another future vacuolated cell at the disc cell stage (n = 0/51 cells quantified from 4 fish). In contrast, future sheath cells almost always had another future sheath cell next to them (n = 221/222 cells quantified from 4 fish).

Altogether, these results establish *jag1a* and Notch activity as the first available markers of vacuolated and sheath cell fates, and confirmed that this cell fate decision is mediated by a lateral inhibition process.

### *her9* and *her6* have a complementary expression pattern to *jag1a*

Having identified that the *jag1a*-Notch alternating pattern correlates with fate, we aimed to identify which are the components of the GRN that make this pattern possible. Notch lateral inhibition model predicts the presence of a Notch target gene that represses *jag1a* expression. This gene should have a mutually exclusive pattern with *jag1a*.

The bHLH genes of the HES/HEY families are good candidates as they are transcriptional repressors often activated by Notch signaling (*Kageyama et al., 2007*). In the notochord, *her9* has been shown to be a Notch target gene (*Yamamoto et al., 2010*). However, the fact that no notochord phenotype was found for the *her9* knockdown zebrafish (*Yamamoto et al., 2010*) suggests functional redundancy with other genes. To identify in an unbiased manner all the HES/HEY genes that repress *jag1a*, we analyzed single-cell RNA-Seq data (*Wagner et al., 2018*). We found that *her6* and *her9* are the

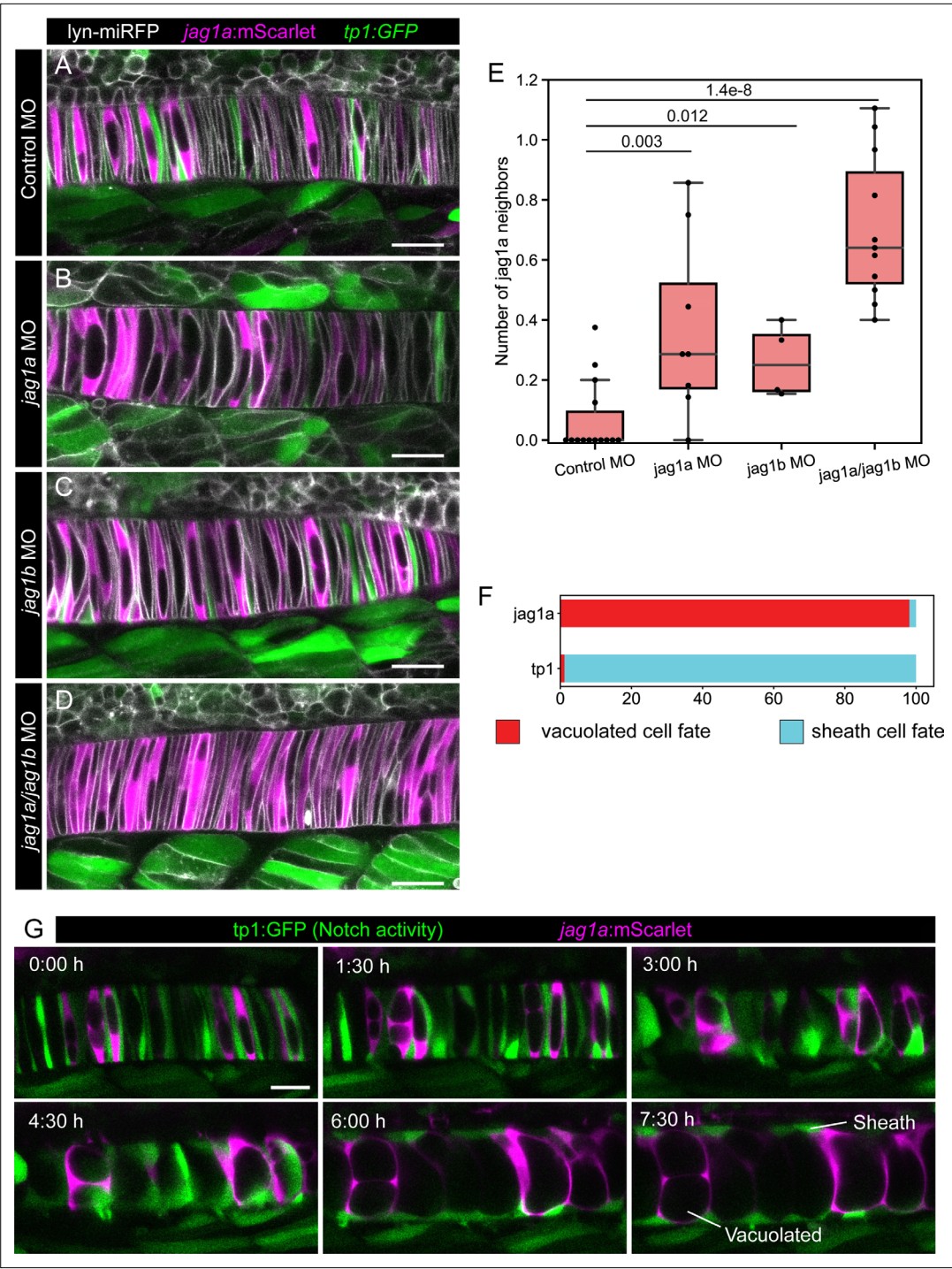

**Figure 2.** *Jag1a/jag1b* inhibition disrupts lateral inhibition pattern, and this pattern correlates with fate. (**A–D**) *jag1a*:mScarlet tp1:GFP 21 hpf fish embryos injected with control MO (**A**), *jag1a* + control MO (**B**), *jag1b* + control MO (**C**), or control MO (**D**), together with lyn-miRFP mRNA to visualize membranes. (**E**) Quantification of the average number of *jag1a*-positive cells directly adjacent to each *jag1a*-positive cell. Two-tailed p-value is shown in the plot. (**F**) Cell fate of cells expressing *jag1a* or the tp1:GFP at the coin-shape stage. Quantifications from images as shown in **G** (standard deviation *jag1a* = 2.696, tp1 = 2.631; n = 5 fish). (**G**) Time lapse of optical sections of notochord cells using the tp1:GFP; *jag1a*:mScarlet double transgenic line. First frame corresponds to 24 hpf. Scale bars, 20μm.

The online version of this article includes the following video and figure supplement(s) for figure 2:

*Figure 2 continued on next page*

*Figure 2 continued*
**Figure supplement 1.** *In vivo* imaging of notochord development.
**Figure 2—video 1.** Time lapse optical section of notochord cells using the tp1:GFP; *jag1a*:mScarlet double transgenic line.
https://elifesciences.org/articles/75429/figures#fig2video1
**Figure 2—video 2.** Maximal projection of the zebrafish notochord optical planes acquired using the feedback microscopy protocol to optimize quality of the region of interest.
https://elifesciences.org/articles/75429/figures#fig2video2
**Figure 2—video 3.** Selected plane from Video S2 stabilizing and magnifying a specific region of the notochord.
https://elifesciences.org/articles/75429/figures#fig2video3

most highly expressed genes of this family in the notochord at 18 and 24 hours post-fertilization (hpf) (*Figure 3A*, *Figure 3—figure supplement 1A-F*). To evaluate their expression pattern, we analyzed mRNA expression by fluorescent in situ hybridization based on a hybridization chain reaction (HCR). *her6* and *her9* were expressed in an alternating pattern with *jag1a* (*Figure 3B–O*). Importantly, in the *her6*/*her9* HCR mRNA staining, we did not observe unlabeled cells, as was the case with tp1, highlighting the importance of identifying endogenous Notch target genes. In contrast to *her6* and *her9* expression, *her12*, which was expressed at a much lower level according to the RNA-Seq, was not detected in the notochord by HCR (*Figure 3—figure supplement 1G-M*). The observed alternating patterns suggest that *her6* and *her9* could repress *jag1a* expression in the notochord.

To analyze if *her6* and *her9* could be direct targets of Notch signaling, we analyzed Rbp-Jκ binding sites in a recently published zebrafish CUT & RUN experiment (*Ye et al., 2021*). Several Rbp-Jκ binding sites were identified in the proximity of *her6* and *her9* transcription start sites, supporting the hypothesis that these genes are direct Notch targets (*Figure 3—figure supplement 1N*, *Ye et al., 2021*).

Aside from the ligand and repressor, the other main component of a lateral inhibition Notch GRN is the Notch receptor. By single-cell RNA-Seq data analysis (*Wagner et al., 2018*) we found that *notch2* was detected in most cells at the highest levels at 18 and 24 hpf (*Figure 3—figure supplement 2A-E*). *notch2* notochord expression was confirmed by fluorescent HCR (*Figure 3—figure supplement 2F–G*). Altogether, we identified the main components of the lateral inhibition GRN, finding *her6* and *her9* as candidate genes to repress *jag1a* expression, and *notch2* as the main Notch receptor in the notochord.

## *her6* and *her9* inhibit *jag1a* expression

To directly assess if *her6* and *her9* are sufficient to inhibit *jag1a* expression, we established notochord-specific genetic mosaics. To that end, we aimed at identifying a highly specific notochord promoter to overexpress *her6* or *her9*, while simultaneously labeling the perturbed cells. Making use of the single-cell RNA-Seq dataset (*Wagner et al., 2018*), we identified *emilin3a* as the gene that offers the best balance between notochord specificity and high expression levels (*Figure 4—figure supplement 1A,B*). We cloned a 5 kb promoter upstream of the coding region and showed that it is sufficient to drive gene expression in the notochord, including most of both *jag1a*:mNeonGreen- and tp1:GFP cells (*Figure 4—figure supplement 1C-J*). Next, we used this promoter and the p2a system (*Kim et al., 2011b*) to generate *her6* or *her9* gain-of-function cells concomitantly with GFP expression, or only-GFP as a control. For each of these constructs, we quantified the level of *jag1a*:mScarlet expression in the GFP-p2a-*her6*, GFP-p2a-*her9* or only-GFP positive cells in comparison to the rest of the notochord. We found that GFP-p2a-*her6* and GFP-p2a-*her9* cells had a lower level of *jag1a*:mScarlet than only-GFP cells, indicating that *her6* and *her9* repress *jag1a* expression in a cell autonomous manner (*Figure 4A–G*). This result was confirmed by quantifying endogenous *jag1a* mRNA expression by fluorescent HCR (*Figure 4—figure supplement 2A-G*).

Having identified *her6* and *her9* as genes sufficient to inhibit *jag1a* expression, we studied if these genes are necessary for lateral inhibition patterning in the notochord. To this end, we generated *her6*/*her9* double transient knockouts (*Figure 3—figure supplement 1N*) in a *jag1a*:mScarlet;*rcn3*:lyn-mNeonGreen background, and quantified the number of *jag1a*-positive cells that are found adjacent to each *jag1a*-positive cell. We found this value to be increased upon *her6* and *her9* gene deletion, showing that *her6* and *her9* are necessary for lateral inhibition (*Figure 4H–J*). Altogether, we show

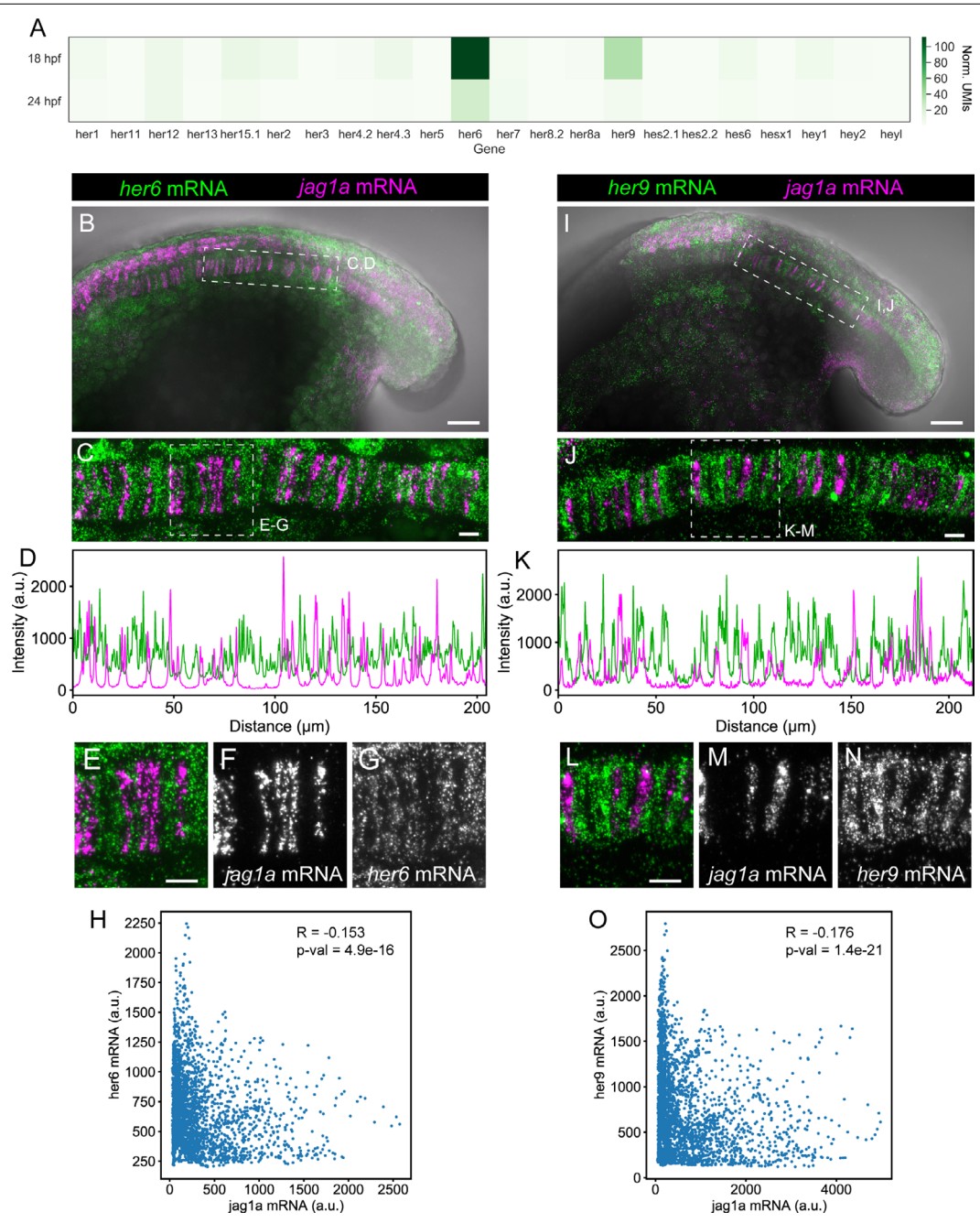

**Figure 3.** *her9* and *her6* show a complementary pattern to *jag1a*. (**A**) Heatmap showing the expression levels of the zebrafish HES/HEY family genes. Values represent average normalized UMIs in all notochord cells at 18 and 24 hpf. (**B**) Projection of confocal optical sections of 18 hpf zebrafish stained with in situ HCR probes against *her6* (green) and *jag1a* (magenta). Transmitted light is shown in gray scale. (**C**) Maximal projection of confocal Airyscan optical sections of the boxed area in (**B**). (**D**), Intensity profile of *her6* (green) and *jag1a* (magenta) along a 1 µm thick horizontal line on the in situ HCR shown in (**C**). (**E–G**) Magnified views of boxed area in (**C**), n = 8. (**H**) Scatter plot of the intensities shown in D. Each point represents *her6* and *jag1a* intensity in a 1-pixel width times 1 um height square. Pearson correlation and p-value of the correlation is shown in the plot. (**I–O**) Analogous images to (**B–H**) based on the *her9* probe instead of *her6* probe, n = 9. Scale bars, 50 µm (**B, I**), 20 µm (**C, E, J, L**).

The online version of this article includes the following figure supplement(s) for figure 3:

**Figure supplement 1.** *her12* expression is not detected in the notochord.

**Figure supplement 2.** Expression of Notch receptors.

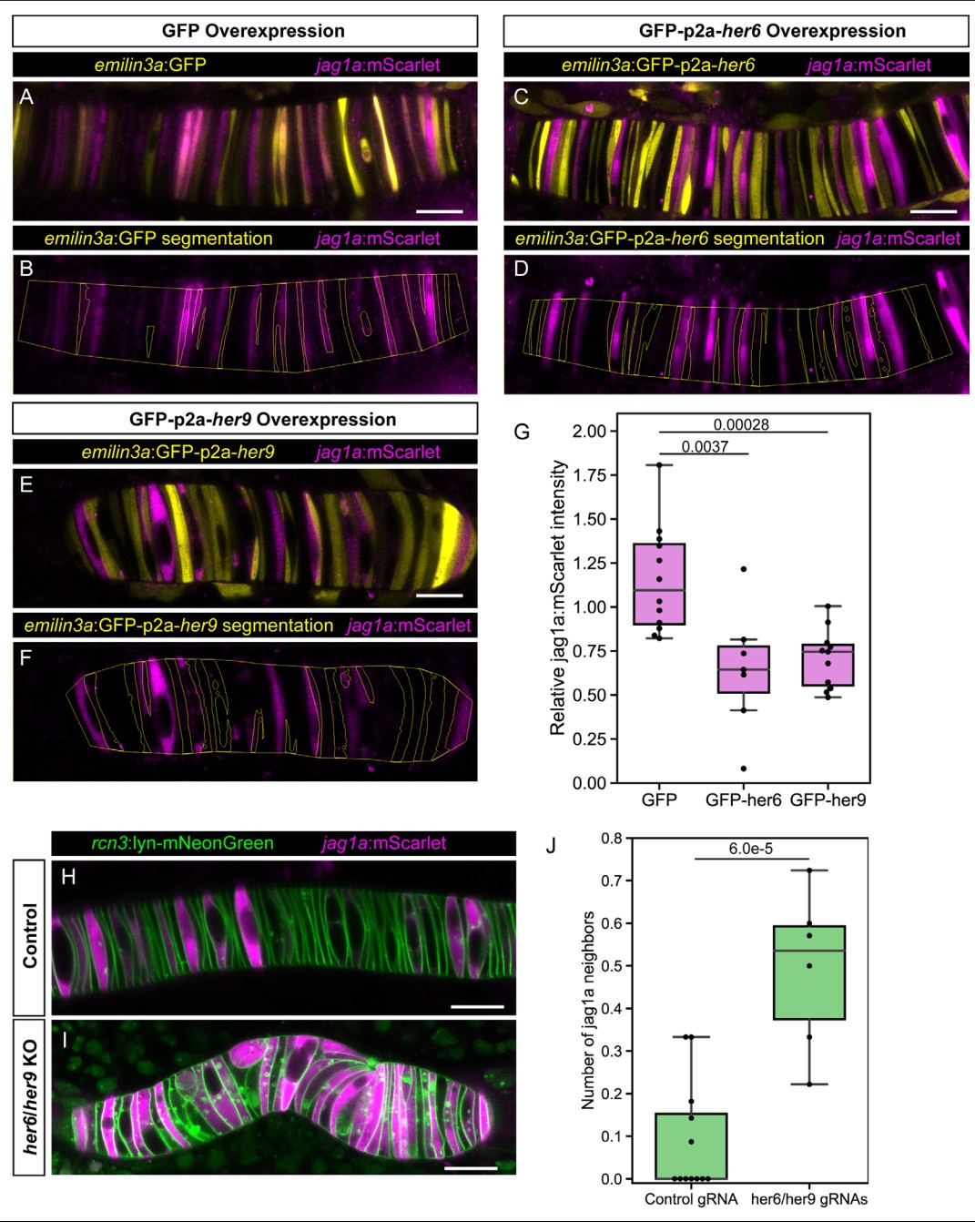

**Figure 4.** *her6* and *her9* inhibit *jag1a* expression. (**A – E**) Airyscan confocal optical sections of live 22 hpf transgenic *jag1a*:mScarlet zebrafish injected with *emilin3a*:GFP (**A and B**), *emilin3a*:GFP-p2a-*her6* (**C and D**) or *emilin3a*:GFP-p2a-*her9* (**E and F**). DNA constructs were injected at the one-cell stage together with I-SceI protein. (**B, D, F**) show the boundary of GFP segmentation in A, C, and E, respectively, and manual outline of the notochord. (**G**), Quantification of *jag1a*:mScarlet intensity inside GFP-positive cells segmented as exemplified in (**B, D, F**). Values in the plot represent the intensity of *jag1a*:mScarlet inside segmented cells divided by the *jag1a*:mScarlet intensity inside the notochord outside of the segmented cells. Each point represents an individual fish (n = 12 GFP, n = 7 GFP-p2a-*her6*, n = 11 GFP-p2a-*her9*). Two-tailed p-values are shown in the plot. (**H**), Airyscan confocal sections of embryo at 22 hpf injected with Cas9 together with a control guide (**H**) or Cas9 together with *her6*/*her9* gRNAs (**I**). (**J**) Quantification of the average number of *jag1a*-positive cells directly adjacent to each *jag1a*-positive cell. For each individual fish, we count how many *jag1a*-positive cells are adjacent to each *jag1a*-postive cell, and then calculate the average for that fish. This value would be equal to 2 in case all the cells are *jag1a*-positive, and zero if no *jag1a*-positive cell is adjacent to another *jag1a*-positive cell. Each individual point

*Figure 4 continued on next page*

*Figure 4 continued*

in the plot represents the average value for an independent fish (n = 7 control, n = 8 *her6*/*her9* KOs). Two-tailed p-value is shown in the plot. Scale bars, 20 μm.

The online version of this article includes the following figure supplement(s) for figure 4:

**Figure supplement 1.** *emilin3a*-5kb promoter drives expression to the notochord.

**Figure supplement 2.** *jag1a* mRNA expression upon *her6* or *her9* overexpression.

that *her6* and *her9* are the repressors in the GRN that generate a lateral inhibition pattern in the notochord.

## *her6*/*her9* and *jag1a* determine notochord cell fate

To test if the identified GRN genes are sufficient to determine cell fate, we first expressed GFP-p2a-*her6*, GFP-p2a-*her9* or only-GFP in a mosaic fashion in the notochord cells, and evaluated its effect on cell fate. At 2 days postfertilization (dpf), a stage where vacuolated and sheath cells can be distinguished, we found a higher proportion of sheath cells in GFP-p2a-*her6* and GFP-p2a-*her9* expressing cells. This result indicates that *her6* and *her9* are sufficient to determine sheath cell fate (*Figure 5A–D*).

Next, we expressed GFP-p2a-*jag1a* or only-GFP. Interestingly, we found that the Notch ligand *jag1a* is sufficient to drive vacuolated cell fate in the same cells where it is expressed (*Figure 5E–G*). Taken together, our results show that not only the Notch targets *her6*/*her9* drive cell fate, but also the Notch ligand *jag1a* determines cell fate in the same cell where it is expressed.

## Stronger cis than trans interactions are required for lateral inhibition patterning

After observing that *jag1a*, a Notch ligand, drives vacuolated cell fate on the same cell where it is expressed, we next investigated the mechanism mediating this process. First, we explored a potential signaling role of the ligand intracellular domain. It has been shown that upon Notch-ligand trans-interaction, not only the NICD is cleaved in the receiver cell, but also the intracellular domain of some ligands, including Jag1, is cleaved inside the sender cell, leading to bidirectional signaling (*Ikeuchi and Sisodia, 2003*; *Kim et al., 2011a*; *Kolev et al., 2005*; *LaVoie and Selkoe, 2003*; *Liebler et al., 2012*; *Metrich et al., 2015*). The intracellular domain of *jag1a* (JICD) would then inhibit Notch signaling in the sender cell (*Kim et al., 2011a*). Thus, overexpression of the full-length ligand in our experiment would increase the amount of ligand that is available to be cleaved, leading to Notch inhibition and promoting vacuolated cell fate. To test this hypothesis, we expressed mScarlet-p2a-JICD or only-mScarlet in a mosaic fashion in notochord cells. We did not observe any effect of JICD on cell fate (*Figure 5—figure supplement 1*), showing that JICD signaling is not sufficient to explain the *jag1a* effect on fate in the notochord.

Next, we considered two different signaling circuits that could explain how *jag1a* can promote vacuolated cell fate in the cells where it is expressed. First, through trans-interactions with the Notch receptor, *jag1a* could activate Notch signaling and as a consequence, *her6*/*her9* expression in their neighbors. *Her6* and *her9* would inhibit *jag1a* in these neighbors, and this would in turn diminish the amount of Notch signaling that the initial cell receives, promoting vacuolated cell fate. A second possible explanation comes from the observation that when Notch ligands are expressed in the same cell as the Notch receptor, they can mutually inhibit each other through cis-inhibition (*Celis de and Bray, 1997*; *Klein et al., 1997*; *Micchelli et al., 1997*). Thus, overexpression of *jag1a* would deplete the Notch receptor in a cell-autonomous manner, making this cell non-responsive to Notch signaling and thus promoting vacuolated cell fate (*Figure 6A*).

To study which of these genetic circuits is predominant in the notochord, we overexpressed *jag1a-GFP* or only-GFP in some notochord cells and quantified *her6* and *her9* expression both within the same cell and in their neighboring cells. We found only a minor or no increase in *her6*/*her9* expression in the neighboring cells (*Figure 6B, D, E, G,I*), suggesting a small Notch-ligand trans-interaction. On the other hand, we observed a strong reduction of *her6*/*her9* expression in the *jag1a*-expressing cells (*Figure 6C, F and H*). Although we cannot rule out that the small effect in the neighboring cells is due to limiting Notch receptor levels, the strong effect observed in the *jag1a*-expressing cells suggests the main mechanism regulating cell fate in its own cell is cis-inhibition.

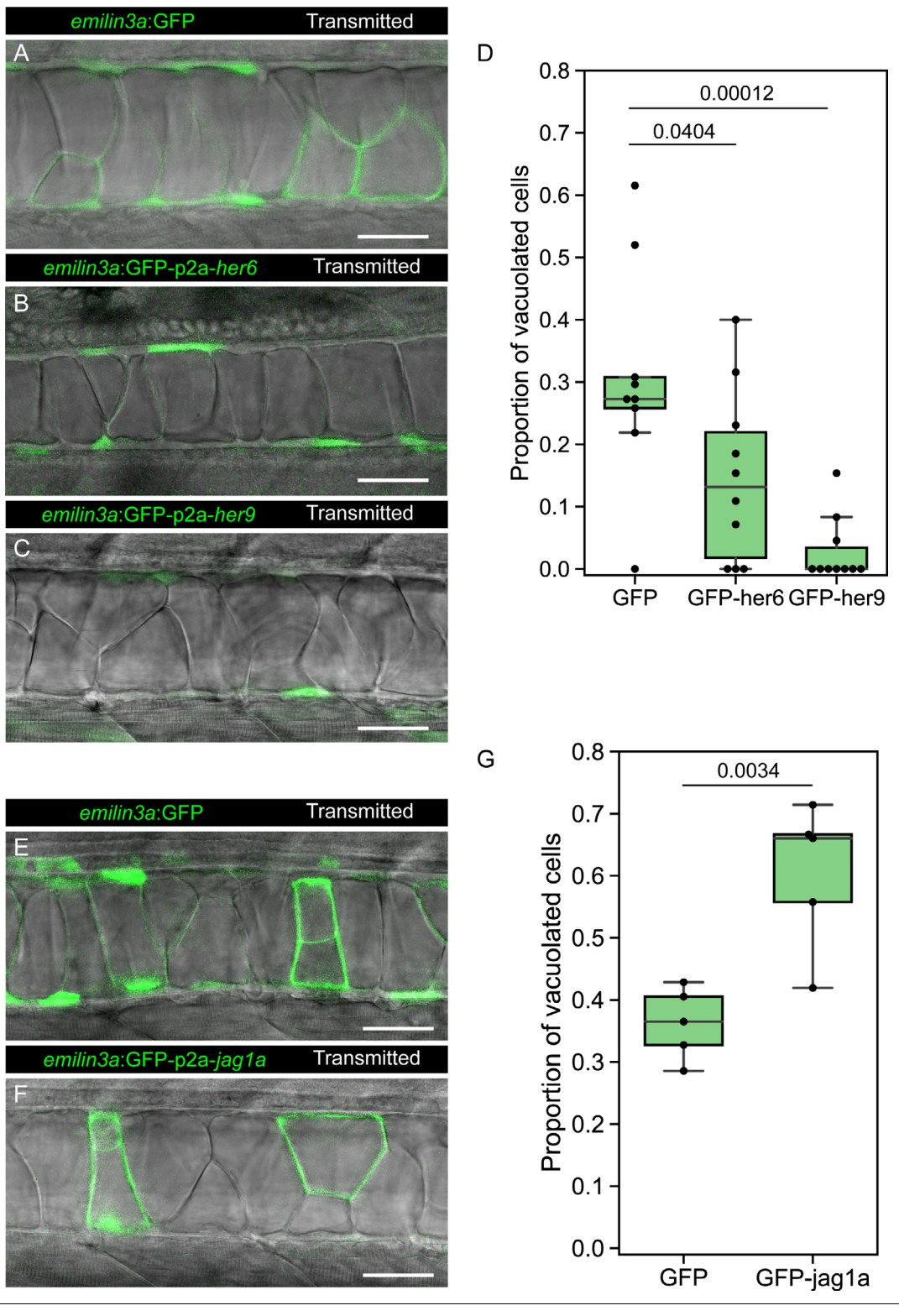

**Figure 5.** *her6*, *her9*, and *jag1a* determine cell fate in the zebrafish notochord. (**A–C, E–F**) Confocal optical sections of 2 dpf live zebrafish that were injected with the *emilin3a*:GFP (**A, E**) *emilin3a*:GFP-p2a-*her6* (**B**), *emilin3a*:GFP-p2a-*her9* (**C**) or *emilin3a*:GFP-p2a-*jag1a* (**F**) constructs. DNA constructs were injected at the one-cell stage together with I-SceI protein. (**D and G**) Proportion of vacuolated cells at 2 dpf are shown. Proportion of vacuolated cells was calculated by counting the number of vacuolated cells divided by the sum of the number of

*Figure 5 continued on next page*

*Figure 5 continued*

sheath and vacuolated cells. Each point in D, G represents an independent fish quantified from on z-stack confocal planes (D, n = 9 GFP, n = 10 GFP-her6, n = 10 GFP-her9, G, n = 5 GFP, n = 5 GFP-jag1a). Two-tailed p-values are shown in D and G. Scale bars, 50 µm.

The online version of this article includes the following figure supplement(s) for figure 5:

**Figure supplement 1.** *Jag1a* intracellular domain does not have an effect on notochord cell fate.

It has previously been shown that cis-interactions are necessary for patterning in absence of cooperativity (*Formosa-Jordan and Ibañes, 2014*; *Sprinzak et al., 2010*; *Sprinzak et al., 2011*). However, how relative values of interaction in cis – within the same cell – and in trans – between neighboring cells – affect patterning has not been explored. Our experimental results suggest a key role of cis-interactions. To better understand which interactions are required for patterning, we implemented a mathematical model that includes ligand-receptor interactions both in cis and in trans based on *Sprinzak et al., 2010*; *Figure 6—figure supplement 1*. Receptor-ligand cis- and trans-interactions are represented by the $K_{cis}$ and $K_{trans}$ parameters, respectively. We next used this model to dissect which combinations of cis- and trans-interactions lead to the lateral inhibition pattern observed experimentally (*Figure 1G–M*). To do so, we evaluated the stability of the homogeneous steady state (HSS) depending on $K_{cis}$ and $K_{trans}$. The HSS is defined as the steady state where all the cells have identical concentrations of Notch ligand, receptor and repressor. When the HSS is stable, the system remains in this homogenous state and no patterning occurs. HSS stability can be evaluated by performing linear stability analysis to calculate the Maximal Lyapunov Exponent (MLE), which represents the exit speed from the homogeneous steady state. Thus, a positive MLE represents an unstable HSS, and this leads to patterning. We found that in the absence of cooperativity, patterning only occurs in a region of the parameter space where $K_{cis}$ is higher than $K_{trans}$ (*Figure 6J*, *Figure 6—figure supplement 2*). If some degree of cooperativity is assumed, patterning is also possible without cis-interactions, as previously described (*Collier et al., 1996*). However, we observed that even in this case, stronger cis- than trans-interactions destabilize the homogeneous state, thus promoting patterning (*Figure 6—figure supplement 2*). These results of our mathematical modeling are in agreement with our experimental observations where we observe a strong cis-inhibition by *jag1a*.

In conclusion, our results show that a *jag1a/jag1b-her6/her9* network generates a lateral inhibition pattern that determines cell fate in the notochord, and that strong ligand-receptor interactions within cells play a key role in the generation of such patterns.

## Discussion

The unidimensional arrangement of cells in the zebrafish notochord, combined with its binary cell fate decisions, make it a unique model to study the properties of the Notch GRN that determines its patterning. One of the most important genetic interactions in a Notch GRN is how the expression of the ligands is regulated by Notch signaling. Previously, it was generally accepted that Notch signaling activates *Jag1* expression leading to lateral induction patterns (*Boareto, 2020*; *Boareto et al., 2015*; *Sjöqvist and Andersson, 2019*). Here we show that Notch signaling, through the activation of the transcriptional repressors *her6* and *her9*, inhibits *jag1a* expression in the notochord, leading to the generation of lateral inhibition patterns. Importantly, *Jag1* is expressed in many other tissues apart from the notochord, including heart, inner ear, muscle, and kidney (*D'Amato et al., 2016*; *Leimeister et al., 2003*; *Lindsell et al., 1995*; *Murata et al., 2006*), suggesting that the identified GRN may be relevant for pattern generation in these other contexts.

Another key part of a Notch GRN that may affect patterning, is whether upon ligand-receptor interaction, there is unidirectional or bidirectional signaling. In the bidirectional signaling situation, not only the cell expressing the receptor would receive a signal, but also the cell expressing the ligand. This signal would be mediated by the intracellular domain (ICD) of the ligand. However, the role of ligand ICDs remains unclear. Previous work showed that the ICD of JAG1 and DLL1 modulate cell differentiation, proliferation, and Notch signaling (*Ikeuchi and Sisodia, 2003*; *Kim et al., 2011a*; *Kolev et al., 2005*; *LaVoie and Selkoe, 2003*; *Metrich et al., 2015*). In contrast, other studies found little or no effect of DLL1-ICD, DLL4-ICD, and JAG1-ICD on gene expression and migration in endothelial cells (*Liebler et al., 2012*). In agreement with the latter, we found no role of the zebrafish *jag1a*-ICD on cell

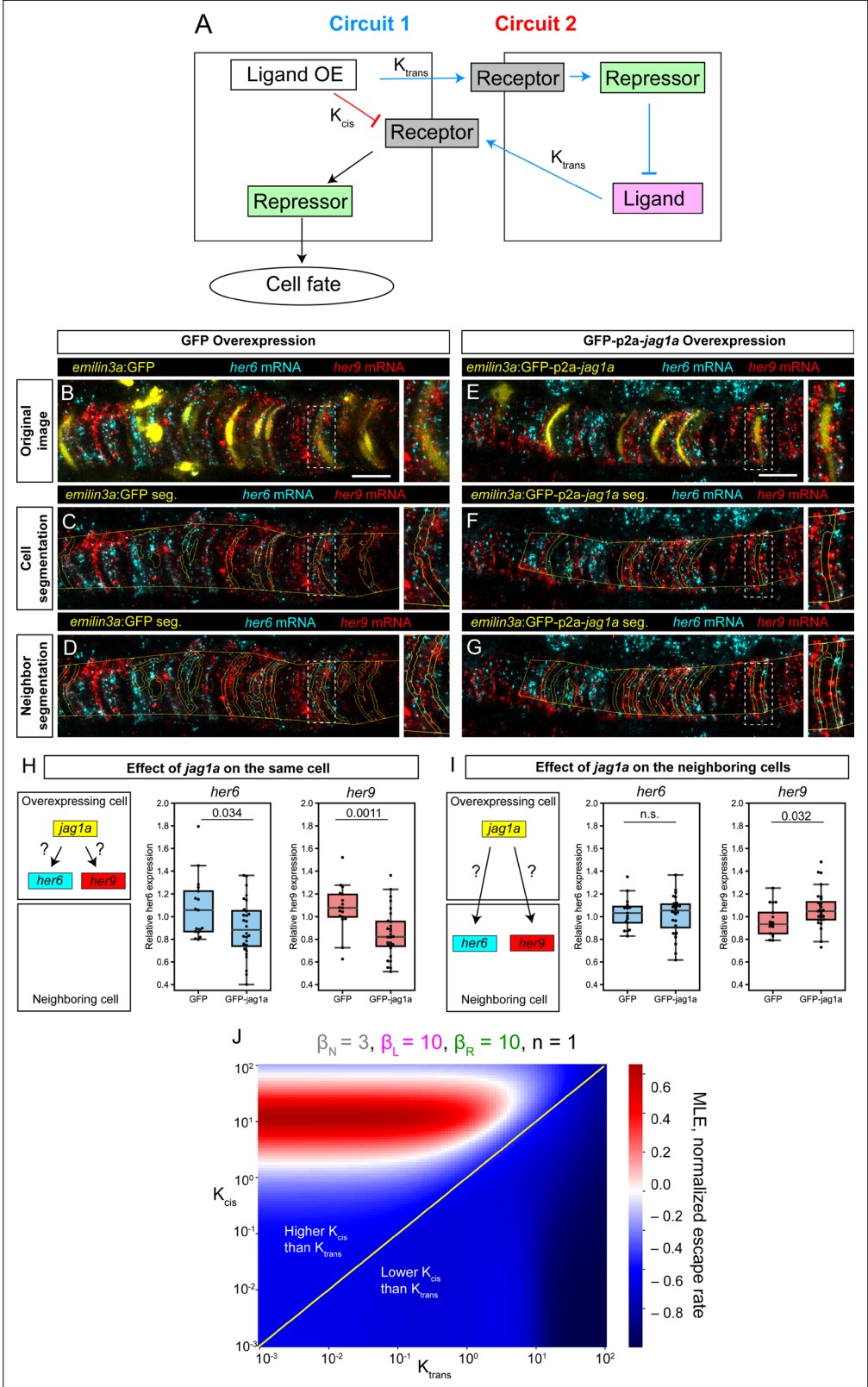

**Figure 6.** Modeling and experimental results of cis- and trans-interactions in the notochord. (**A**) Two possible circuits may explain the effect of *jag1a* on fate of the cell where *jag1a* is overexpressed. Circuit 1 is based on the interaction of ligand and receptor in trans. Circuit 2 is based on a possible role of cis-inhibition of the Notch receptor by the ligand. Cells where we overexpress the ligand are represented as the cell on the left. Adjacent

*Figure 6 continued*

cells are represented on the right. OE, overexpression. (**B – G**), Airyscan confocal planes of fixed 22 hpf transgenic fish injected with *emilin3a*:GFP (**B–D**) or *emilin3a*:GFP-p2a-*jag1a* (**E–G**) constructs. GFP was detected by antibody staining and *her6* and *her9* mRNA by in situ HCR in whole mount embryos. (**C and F**) show the notochord outline manually selected and the outline of GFP-positive cells automatically segmented. (**D and G**) show the outline of the manually selected notochord and the neighborhood to the GFP-positive cells. On the right side of each panel, a magnified view of the boxed region is shown. (**H, I**) Quantification of *her6* and *her9* mRNA expression after GFP-based segmentation as shown in (**C, F**) or (**D, G**), respectively. Values of *her6* and *her9* expression levels inside the segmented area inside the notochord were divided by the expression levels of the same genes in the region outside the segmented area, also inside the notochord. Each point represents a different fish. Two-tailed p-values are shown in the plots. n.s., non-significant. (**J**) Escape rates from the homogeneous steady state (indicated by Maximum Lyapunov Exponents, or MLE) as a function of $K_{cis}$ and $K_{trans}$ parameters. Positive MLE values (red) support patterning, while negative MLE values (blue) do not. Scale bars, 20 μm.

The online version of this article includes the following figure supplement(s) for figure 6:

**Figure supplement 1.** Model of lateral inhibition including cis-interactions.

**Figure supplement 2.** Escape rates from the homogeneous steady state.

---

fate. Further research will be needed to elucidate if the role of ligand ICDs depends on the signaling context, and whether different cell types respond differently to ICDs.

Patterning not only depends on the topology of a GRN, but also on the strength of each of the interactions. Here, using mathematical simulations supported by experimental results, we shed light on which combinations of parameters promote pattern generation. Specifically, we find that a stronger Notch-ligand interaction in cis than in trans is key for pattern generation. Importantly, this does not mean that trans-interactions are not needed. In absence of such interactions, there would be no communication between cells and thus no lateral inhibition patterning.

The strength and signaling efficiency of cis- and trans-interactions in Notch GRNs depend on the specific ligand-receptor pair (*Benedito et al., 2009*; *Luca et al., 2015*; *Petrovic et al., 2014*; *Sjöqvist and Andersson, 2019*). Some DLLs, such as DLL4, activate Notch signaling in trans more strongly than Jagged ligands (*Benedito et al., 2009*). On the other hand, the *Drosophila* homolog of Jagged genes, *serrate*, inhibits Notch receptors in cis more efficiently than Delta ligands (*de Celis and Bray, 2000*; *del Álamo et al., 2011*; *Klein et al., 1997*; *Li and Baker, 2004*). The possibilities of imaging and genetic manipulation that the zebrafish offers, together with the unique cell-cell contacts in the notochord, will make this organ a very valuable *in vivo* system to evaluate the properties not only of endogenous ligands, but also other Notch ligands, to better understand how cis and trans parameters determine pattern generation.

Our results not only explain how Notch drives pattern generation, but also how cell fate is determined during notochord development. We identified Notch activity and its downstream genes *her6* and *her9* as key determinants of sheath cell fate in the notochord. In some tissues, including skeletal muscle, intestine and neural systems, a higher Notch activity is related to stemness, while a lower Notch activity is related to differentiation (*Blanpain et al., 2006*; *Fre et al., 2005*; *Imayoshi et al., 2010*; *Schuster-Gossler et al., 2007*; *Vasyutina et al., 2007*). This raises the interesting hypothesis of whether sheath cells can be considered as only partially differentiated notochord cells. In agreement with this concept is the recent finding that upon vacuolated cell damage, sheath cells develop vacuoles and partially restore notochord structure (*Garcia et al., 2017*; *Lopez-Baez et al., 2018*). However, a possible role of Notch signaling during notochord regeneration is yet to be tested.

Several pieces of evidence suggest that the GRN that we have identified is not exclusive to zebrafish. Previous studies based on BAC transgenesis showed that *Hes1*, the mammalian homolog of *her6* and *her9*, is expressed in the mouse notochord, suggesting it may play a role in the patterning of the mammalian notochord (*Klinck et al., 2011*). Problems in notochord development have been associated with defects in spine morphogenesis (*Bagwell et al., 2020*; *Gray et al., 2014*; *Gray et al., 2021*; *Sun et al., 2020*). Interestingly, mutations in JAG1 and NOTCH2 (*McDaniell et al., 2006*; *Oda et al., 1997*), the human homologs of the main ligands and receptor in the zebrafish notochord, lead to vertebrae malformations in human Alagille Syndrome. This suggests that spine problems in this human syndrome may be the result of

defective Notch patterning during notochord development. Thus, in this study, we describe a GRN that is likely conserved across vertebrates, opening the door to better understand how mutations in *JAG1* or *NOTCH2* lead to the problems observed in the human disease.

In non-vertebrate chordates such as ascidians, a single cell type performs the two main functions of both sheath cells and vacuolated cells: covering the surface and producing the fluid (*Deng et al., 2013*; *Dong et al., 2009*). From an evolutionary perspective, it is plausible that Notch signaling was involved in dividing these possible ancestral functions into two different cell types. We speculate that Notch- or Hes-responsive enhancers were co-opted during vertebrate evolution to control the expression of the key genes necessary for vacuolated and sheath cell functions, making the specialization of the two different cell types possible. Given how frequently Notch signaling determines cell fate across development, Notch could represent a general mechanism that facilitated division of functions between different cells, promoting the evolution of new cell types.

Altogether, we have established the notochord as a new model system to study the principles that determine pattern generation. Using a combination of mathematical modeling, single-cell RNA-Seq analysis and genetic perturbation approaches, we identified *jag1a*, *her6*, *her9* and *notch2* as the key genes that determine cell fate and patterning. We expect that the GRN properties identified in this study will help understand the principles underlying patterning and cell fate decisions across multicellular organisms.

# Materials and methods

## Key resources table

| Reagent type (species) or resource | Designation | Source or reference | Identifiers | Additional information |
|---|---|---|---|---|
| Antibody | GFP-Booster Alexa Fluor 488. anti-GFP Alpaca/recombinant V$_H$H domain, monoclonal. | Chromotek | gb2AF488 | Used at (1:500) dilution. |
| Genetic reagent (*Danio rerio*) | *jag1a*:mScarlet | This paper | | "Animal handling and generation of transgenic lines" Methods section |
| Genetic reagent (*Danio rerio*) | *jag1a*:mNeonGreen | This paper | | "Animal handling and generation of transgenic lines" Methods section |
| Genetic reagent (*Danio rerio*) | *emilin3a*:mScarlet | This paper | | "Animal handling and generation of transgenic lines" Methods section |
| Genetic reagent (*Danio rerio*) | *rcn3*:lyn-mNeonGreen | This paper | | "Animal handling and generation of transgenic lines" Methods section |
| Genetic reagent (*Danio rerio*) | tp1:GFP | *Parsons et al., 2009* | | "Animal handling and generation of transgenic lines" Methods section |
| Sequence-based reagent | Primer 1: pTarBAC_HA1_iTol2_F | This paper | Primer | gcgtaagcggggcacatttcattacctctttctccgcacccgacatagatCCCTGCTCGAGCCGGGCCCAAGTG |
| Sequence-based reagent | Primer 2: pTarBAC_HA2_iTol2_R | This paper | Primer | gcggggcatgactattggcgcgccggatcgatccttaattaagtctactaATTATGATCCTCTAGATCAGATC |
| Sequence-based reagent | Primer 3: jag1a_HA1_mScarlet_F | This paper | Primer | gaggcgtgtggcggctgaagtggtagttttcacagcgacagacacacagacagacaaaccACCATGGTGAGCAAGGGC |
| Sequence-based reagent | Primer 4: jag1a_HA2_FRT_R | This paper | Primer | agcagcacgtgagcggacagcgccgcaaaagttgagctcggtctgagaatGGAGGCTACCATGGAGAAG |
| Sequence-based reagent | Primer 5: jag1a_HA1_mNG_F | This paper | Primer | gaggcgtgtggcggctgaagtggtagttttcacagcgacagacacacagacagacaaaccACCATGGTGAGCAAGGGC |
| Sequence-based reagent | Primer 6: Scaffold | *Shah et al., 2015* | Primer | GATCCGCACCGACTCGGTGCCACTTTTTCAAGTTGATAACGGACTAGCCTTATTTTAACTTGCTATTTCTAGCTCTAAAAC |
| Sequence-based reagent | Primer 7: her6_guide1 | This paper | Primer | taatacgactcactataGGTGGTCGGCGCCCCTCCATgttttagagctagaa |
| Sequence-based reagent | Primer 8: her6_guide2 | This paper | Primer | taatacgactcactataGGGTGGCCATTCTTTGAAGGgttttagagctagaa |
| Sequence-based reagent | Primer 9: her9_guide1 | This paper | Primer | taatacgactcactataGGGTGACTGACAGCCCGCGGgttttagagctagaa |
| Sequence-based reagent | Primer 10: her9_guide2 | This paper | Primer | taatacgactcactataGGGGGAAACCCTGCGGCCGTgttttagagctagaa |

*Continued on next page*

*Continued*

| Reagent type (species) or resource | Designation | Source or reference | Identifiers | Additional information |
|---|---|---|---|---|
| Sequence-based reagent | Primer 11: univ_guide | *Wierson et al., 2020* | Primer | taatacgactcactataGGGAGGCGTTCGGGCCCACAGgttttagagctagaa |
| Sequence-based reagent | Primer 12: her6_F | This paper | Primer | GTTTGCTGTTTCTGAGCGGAG |
| Sequence-based reagent | Primer 13: her6_R | This paper | Primer | GGGAAGCACGTCTGAGTCTG |
| Sequence-based reagent | Primer 14: her9_F | This paper | Primer | CCGCGCAGTATGTGAATGC |
| Sequence-based reagent | Primer 15: her9_R | This paper | Primer | ACCTTCACAGGCTACAGAACC |
| Sequence-based reagent | Control MO | *Yamamoto et al., 2010* | Morpholino | CCTCTTACCTCAGTTACAATTTATA |
| Sequence-based reagent | jag1a_splMO | *Yamamoto et al., 2010* | Morpholino | AAGCCAAACCCGCACATACCCGCAT |
| Sequence-based reagent | jag1b_atgMO | *Yamamoto et al., 2010* | Morpholino | CTGAACTCCGTCGCAGAATCATGCC |
| Recombinant DNA reagent | mScarlet FRT kan FRT | This paper | | Sequence available in *Source data 1* file. |
| Recombinant DNA reagent | mNG FRT kan FRT | This paper | | Sequence available in *Source data 2* file. |
| Recombinant DNA reagent | emilin3a mScarlet | This paper | | Sequence available in *Source data 3* file. |
| Recombinant DNA reagent | rcn3 lyn mNeonGreen | This paper | | Sequence available in *Source data 4* file. |
| Recombinant DNA reagent | SP6 lyn-miRFP-pA | This paper | | Sequence available in *Source data 5* file. |
| Recombinant DNA reagent | emilin3a GFP | This paper | | Sequence available in *Source data 6* file. |
| Recombinant DNA reagent | emilin3a GFP-p2a-her6 | This paper | | Sequence available in *Source data 7* file. |
| Recombinant DNA reagent | emilin3a GFP-p2a-her9 | This paper | | Sequence available in *Source data 8* file. |
| Recombinant DNA reagent | emilin3a mScarlet-p2a-JICD | This paper | | Sequence available in *Source data 9* file. |
| Commercial assay or kit | mMESSAGEmMACHINESP6 Transcription Kit | Thermo Fisher Scientific | Cat#:AM1340 | |
| Commercial assay or kit | Tricaine (MESAB) | Sigma-Aldrich | Cat#:A5040 | |
| Software, algorithm | Code for image analysis and mathematical modeling | This paper | | Available in github: https://github.com/hsancheziranzo/notochord-lateral-inhibition (copy archived at swh:1:rev:2e5c5fe15e30ea6bacdc0282e1506b44b05415af, *Sánchez-Iranzo, 2022*) |

## Animal handling and generation of transgenic lines

The construct to generate Tg(*jag1a*:mScarlet) transgenic line was generated by BAC recombineering using the CH211-21D8 BAC. We first used EL250 (*Lee et al., 2001*) bacteria to recombine first the iTol2Amp cassette (*Suster et al., 2011*, primers 1 and 2, Key Resources Table) and substitute the loxP site in the BAC backbone. To recombine the mScarlet sequence into the BAC, we first used Gibson Assembly to substitute mCherry-p2a-CreERT2 by mScarlet in the mCherry-p2a-CreERT2-FRT-kan-FRT plasmid (*Sánchez-Iranzo et al., 2018*) to generate an mScarlet-FRT-kan-FRT plasmid (*Source data 1*). Then, we used the primers 3 and 4 (Key Resources Table) to amplify and recombine the mScarlet-FRT-kan-FRT into the ATG of *jag1a* in the BAC CH211-21D8. Finally, we removed the kanamycin resistance by activating flipase expression in the EL250 bacteria.

Similarly, we generated the jag1a:mNeonGreen BAC by first using Gibson Assembly to generate the mNeon-Green-FRT-kan-FRT plasmid (*Source data 2*). Next, we used primers 4 and 5 (Key Resources Table) to amplify the mNeonGreen-FRT-kan-FRT into the ATG of the *jag1a* BAC, followed by kanamycin resistance removal.

To clone the *emilin3a*:mScarlet plasmid (*Source data 3*) we selected the 5 kb upstream of the *emilin3a* ATG and cloned it upstream of mScarlet in a tol2 plasmid. The *rcn3*:lyn-mNeonGreen construct (*Source data 4*) was generated by Gibson Assembly using the previously described *rcn3* promoter (*Ellis et al., 2013*).

*jag1a*:mScarlet, *jag1a*:mNeonGreen, *emilin3a*:mScarlet and *rcn3*:lyn-mNeonGreen were injected at the one cell stage using tol2 transposase. To establish the stable transgenic lines, we crossed the fish by wild type until we found 50% of the progeny transgenic, indicative of a probable single insertion. For the *rcn3*:mNeonGreen transgenic line, due to the high variability in gene expression between different lines, we selected the most notochord specific line among 5–10 different founders.

As a reporter of Notch activity, we used the tp1:GFP line (*Parsons et al., 2009*). This line includes six copies of the promoter from the Epstein-Barr Virus terminal protein 1 (TP1), cloned upstream of the rabbit β-globin minimal promoter. Each TP1 copy contains two Rbp-Jκ binding sites.

All experiments were performed on embryos younger than 3 dpf, as is stipulated by the EMBL internal policy 65 (IP65) and European Union Directive 2010/63/EU.

## *her6* and *her9* Knockout

To generate *her6* and *her9* transient knockout (crispants), we designed guide RNAs (gRNAs) targeting the beginning and the end of both *her6* and *her9*, resulting in whole gene deletion. Guides were identified using CRISPRscan (*Doench et al., 2014*; *Moreno-Mateos et al., 2015*) and synthesized as previously described (*Shah et al., 2015*; Primers 6–10, Key Resources Table). The injection mix included custom-produced Cas9-GFP at 2.4 mg/mL, KCl 300 mM and the four gRNAs, each of them at 12.5 ng/μL. Only embryos where the antero-posterior axis was shortened were selected for imaging. As a control, we used embryos where a gRNA with no target in the zebrafish genome (*Wierson et al., 2020*; Primer 11, Key Resources Table) was injected. Primers 12–15 (Table S2) were used for the detection of the deleted allele in all the fish used for imaging. Effective deletion was confirmed by sequencing of two KO *her6* and two KO *her9* PCR products; only embryos where both a *her6* and *her9* knockout band was detected by PCR (7/10) were considered for the quantification. Heterozygous embryos for both *rcn3*:mNeonGreen and *jag1a*:mScarlet transgenes were used in this experiment. Cells with *jag1a*:mScarlet intensity lower than 10% of the maximum intensity value in each image were considered negative for *jag1a*.

## *jag1a* and *jag1b* MOs

The injection mix contained 100 ng/uL of lyn-miRFP mRNA and 0.4 mM of MO (Gene Tools). Specifically, the *jag1a/jag1b* mix contained 0.2 mM *jag1a* + 0.2 mM *jag1b*, the *jag1a* mix contained 0.2 mM *jag1a* + 0.2 control MO, the *jag1b* mix contained 0.2 mM *jag1b* + 0.2 mM control MO, and the control MO mix contained 0.4 mM of control MO. *jag1a* and *jag1b* MOs had been described and validated previously (*Yamamoto et al., 2010*).

mRNA was generated by digestion of the SP6 lyn-miRFP-pA plasmid (*Source data 5*) with NotI, followed by SP6 mediated transcription (mMessage mMachine SP6, Thermo Fisher Scientific).

The lyn-miRFP (*Shcherbakova et al., 2016*) mRNA injected, not only allowed membrane labeling, but it is also a control of injection. Few embryos where the infrared membrane signal was not detected were excluded from the analysis. Cells with *jag1a*:mScarlet intensity lower than 10% of the maximum intensity value in each image were considered negative for *jag1a*.

Cell fate analysis *emilin3a*:GFP (*Source data 6*), *emilin3a*:mScarlet (*Source data 3*), *emilin3a*:GFP-p2a-*her6* (*Source data 7*), *emilin3a*:GFP-p2a-*her9* (*Source data 8*) or *emilin3a*:mScarlet-p2a-*jag1a* (*Source data 9*) were cloned using Gibson Assembly using as template synthesized *her6*, *her9*, and *jag1a* cDNAs. These plasmids were injected at the one cell stage using Isce-I as previously described (*Rembold et al., 2006*). GFP fluorescence and transmitted light were imaged *in vivo* at 2 dpf. Quantifications were made on 3D confocal stacks. Number of cells were manually quantified using the Cell Counter Fiji plugin (*Schindelin et al., 2012*).

## Hybridization chain reaction and immunofluorescence

First, *emilin3a*:GFP, *emilin3a*:GFP-p2a-*her6*, *emilin3a*:GFP-p2a-*her9* or *emilin3a*:mScarlet-p2a-*jag1a* constructs were injected at the one cell stage and fish were fixed at 20–22 hpf. Hybridization chain reaction (Molecular Instruments) was performed following manufacturer instructions. *her6*, *her9*,

jag1a, jag1b and notch2 probes were produced by Molecular Instruments as 20 probe set sizes. If GFP needed to be detected, after HCR protocol, samples were incubated overnight with anti-GFP nanobody A488 (gb2AF488, Chromotek, 1:500), followed by 5 × 30 min SSCT 5 X washing steps.

## Single-cell RNA-Seq analysis

Single-cell RNA-Seq data was obtained from *Wagner et al., 2018* (*Wagner et al., 2018*). We filtered the raw data and selected the cells labeled as notochord in the original publication, and analyzed them using the Scanpy v1.4.4 (*Wolf et al., 2018*) python package. UMAP coordinates were calculated using normalized non-logarithmically transformed values and the scanpy.pp.neighbors function with n_neighbors = 20 and n_pcs = 5 parameter values. log(UMI +1) values were represented in the UMAP plots, where log represents natural logarithm. Boxplots and heatmaps were generated using the seaborn python package.

*emilin3a* was found as the gene with the best balance between notochord enrichment and high expression levels. We did this by selecting the gene with the highest score according to this equation:

$$score = \frac{Expr\_N^2}{Expr\_NN} \qquad (1)$$

where *Expr_N* represents the average of normalized UMIs for each gene across all notochord cells at 18 hpf, and *Expr_NN* represents the analogous values for the non-notochord cells at the same stage. Genes with the highest score are shown in *Table 1*.

Interpretation of the data was supported by the extensive data available in ZFIN (*Howe et al., 2021*).

**Table 1.** Genes with a highest score for specificity and expression levels in the notochord at 18 hpf.

Expression: Average expression in Notochord cells (normalized UMIs per million). Enrichment: Average expression in notochord cells divided by average expression in the rest of the cells in the fish at 18 hpf. Score: Expression multiplied by enrichment (equivalent to the equation described above).

| Gene | Expression | Enrichment | Score |
|---|---|---|---|
| emilin3a | 5055.75 | 1118.16 | 5653.16 |
| ntd5 | 5929.62 | 134.03 | 794.74 |
| col2a1a | 9083.50 | 67.75 | 633.59 |
| cmn | 1536.94 | 330.67 | 508.22 |
| loxl5b | 1264.29 | 330.84 | 418.28 |
| col9a1b | 1511.20 | 226.72 | 342.62 |
| ta | 1443.23 | 162.65 | 234.74 |
| LOC100333762 | 642.86 | 303.86 | 195.34 |
| lgals1l1 | 2163.88 | 86.59 | 187.38 |
| col9a2 | 1331.86 | 103.37 | 137.68 |
| si:ch211-125-g7.4 | 501.98 | 272.60 | 136.84 |
| si:dkey-12l12.1 | 408.15 | 304.23 | 127.17 |
| col9a3 | 754.32 | 140.39 | 105.90 |
| LOC100334188 | 393.62 | 205.68 | 80.96 |
| pmp22b | 1001.80 | 68.64 | 68.76 |
| si:dkey-99l1.9 | 499.03 | 131.45 | 65.60 |
| si:ch73-23l24.1 | 317.78 | 205.60 | 65.34 |
| lgals2a | 1018.17 | 64.08 | 65.24 |
| twist2 | 306.83 | 167.20 | 51.60 |
| lect1 | 616.97 | 79.24 | 48.89 |

## Electron microscopy

For EM imaging, samples were chemically fixed by immersing them in 2.5% glutaraldehyde and 4% paraformaldehyde in 0.1 M PHEM buffer. Sections were post-stained with uranyl acetate for 5 min and with lead citrate for 2 min. The overall EM protocol is similar to previously reported (*Schieber et al., 2010*).

## Microscopy

Zebrafish embryos were embedded in 0.6% agarose low gelling temperature (A0701, Sigma) with 0.16 mg ml−1 Tricaine in E3 medium. For imaging embryos between 18 and 24 hpf, agarose covering the tail was removed to allow freely development of their tail. Imaging was performed with a Zeiss LSM880 laser scanning confocal microscope, using a 40 x/1.1NA water-immersion objective.

## Adaptive feedback microscopy workflow

The adaptive feedback microscopy workflow was set up on Zeiss LSM880 AiryScan Fast microscope. Automated image analysis and definition of high-zoom tile positions was implemented as a Fiji plugin using previously developed AutoMicTools library (https://git.embl.de/halavaty/AutoMicTools). MyPic VBA macro (*Politi et al., 2018*) was used as a communication interface between the Fiji plugin and ZenBlack software controlling the microscope.

Both low-zoom and high-zoom images were acquired using AiryFast modality to enable time resolution of 5 min. 488 nm line of the Argon laser was used for excitation, fluorescent signal was detected using 499–553 nm emission filter. Low-zoom images were acquired using lowest possible zoom and rectangular tilescan in the total area 991 by 673 µm with the pixel size 0.835 µm and spacing between slices 5 µm. Each high-zoom tile was acquired in the field of view 83.72 by 83.72 µm with the pixel size 0.108 µm and spacing between slices 2.5 µm. Collected high-zoom tiles were stitched in Fiji using BigStitcher plugin (*Hörl et al., 2019*) and custom Jython scripts. To show the same region of the notochord independently on the move of the developing zebrafish, we used a custom-made Fiji Macro where the region of interest was manually selected every 10 frames, and interpolated for the rest of the timepoints.

To show the same region of the notochord independently on the move of the developing embryo, we used a custom-made Fiji Macro where the region of interest was manually selected every 10 frames, and the region of interest interpolated for the rest of the timepoints.

## Image analysis

Python 3.7.4 was used for image analysis. First, the intensities of each of the channels was normalized between 0 and 1, where 0 was assigned to the minimum intensity value in the image, and 1 to the maximum value. Then, a gaussian filter was applied to the channel. This was done using the filters. gaussian_filter function of scipy.ndimage package, with a sigma value equal to 3. Then, both adaptive and global single-value segmentation were applied to the GFP channel. For the global single-value segmentation, the value was chosen automatically for each image as 1.5 times the median intensity of the GFP channel. To generate the adaptive segmentation, we calculated the local mean using as a kernel a uniform circle of 120 pixel diameter, and the rank.mean function of the skimage.filters package. Only the pixels with a higher value than both the global and the adaptive thresholds were considered for further analysis (Segmentation 1).

To define the GFP-positive cells, we filled holes in the cells by applying a 5-iteration binary dilation followed by a 9-interation binary erosion (scipy.ndimage python package). A higher erosion than dilation was applied to avoid defining as GFP-positive cells the pixels in the boundaries between cells. Only objects with an area of 3500 squared pixels were defined as cells and considered for further analysis (Segmentation 2).

The neighborhood of GFP cells was defined as follows. We first applied an 8-pixel binary dilation of 8 pixels to the GFP cells as defined in 'Segmentation 1' to define the boundary between cells. We then applied a 25-pixel binary dilation to define the neighboring cells. The region generated by the 25-pixel dilatation is the region that we considered as 'neighboring cells' (Segmentation 3).

To determine the relative intensity inside the 'GFP-positive cells' or the 'neighboring to GFP cells' we manually selected the notochord region, and we only considered the pixels inside the manually selected region. Then, we measured the mean value of the different mRNA signals inside the selected cells relative to the value of all the notochord.

In all the analyzed images, the stepsize is 63.7 nm/pixel. Plots were generated using boxplot and swarmplot functions of the seaborn python package.

## Statistical analysis

Statistical analysis was performed using the scipy.stats python package. The specific statistical test used, including sample size and the p-values are indicated in the figures and figure legends.

## Data and code availability

Code is available under the MIT open source license on GitHub at: https://github.com/hsancheziranzo/notochord-lateral-inhibition (copy archived at swh:1:rev:2e5c5fe15e30ea6bacdc0282e1506b44b05415af)

(*Sánchez-Iranzo et al., 2021*; *Sánchez-Iranzo, 2022*). Images used for image analysis are available in Mendeley Data: https://doi.org/10.17632/fzmk5k982j.1 (CC BY 4.0).

## Materials availability

Requests for experimental resources and reagents should be directed to and will be fulfilled by Alba Diz-Muñoz (diz@embl.de) or Héctor Sánchez-Iranzo (hector.sanchez@kit.edu).

## Acknowledgements

We thank Anna Erzberger, Aissam Ikmi and Stefano de Renzis for critical reading of the manuscript. We thank Jonas Hartmann for discussion on the project and training on image analysis. We are grateful to the EMBL EM core facility (EMCF), and in particular to Rachel Mellwig and Yannick Schwab, for the EM experiments. We thank the EMBL Fish Facility, and in particular to Sabine Görgens. We thank the EMBL Advanced Light Microscopy Facility, and especially Christian Tischer and Stefan Terjung for image analysis and microscopy support. We thank Alexander Ernst for the development of the custom-made ImageJ macro used to generate some of the movies of the paper. We thank the Life Science Editors for editorial support. Funding: This study was funded by the European Molecular Biology Laboratory (EMBL) and Deutsche Forschungsgemeinschaft (DFG) grants DI 2205/3–1 and DI 2205/2–1 to AD-M. H S-I was funded by the EMBO fellowship (ALTF 306–2018) and the Joachim Herz Stiftung Add-on Fellowship for Interdisciplinary Science.

## Additional information

### Funding

| Funder | Grant reference number | Author |
| --- | --- | --- |
| EMBO | ALTF 306-2018 | Héctor Sánchez-Iranzo |
| Deutsche Forschungsgemeinschaft | DI 2205/3-1 | Alba Diz-Muñoz |
| Deutsche Forschungsgemeinschaft | DI 2205/2-1 | Alba Diz-Muñoz |
| European Molecular Biology Laboratory | | Alba Diz-Muñoz |
| Joachim Herz Stiftung | Add-on Fellowship for Interdisciplinary Life Science | Héctor Sánchez-Iranzo |

The funders had no role in study design, data collection and interpretation, or the decision to submit the work for publication.

### Author contributions

Héctor Sánchez-Iranzo, Conceptualization, Funding acquisition, Investigation, Methodology, Software, Visualization, Writing – original draft, Writing – review and editing; Aliaksandr Halavatyi, Methodology, Software, Visualization; Alba Diz-Muñoz, Conceptualization, Funding acquisition, Supervision, Writing – original draft, Writing – review and editing

### Author ORCIDs

Héctor Sánchez-Iranzo http://orcid.org/0000-0003-2032-0231
Alba Diz-Muñoz http://orcid.org/0000-0001-6864-8901

### Ethics

All experiments were performed on embryos younger than 3 dpf, as is stipulated by the EMBL internal policy 65 (IP65) and the European Union Directive 2010/63/EU.

### Decision letter and Author response

Decision letter https://doi.org/10.7554/eLife.75429.sa1
Author response https://doi.org/10.7554/eLife.75429.sa2

## Additional files

### Supplementary files
- MDAR checklist
- Source data 1. mScarlet FRT kan FRT.
- Source data 2. mNG FRT kan FRT.
- Source data 3. emilin3a mScarlet.
- Source data 4. rcn3 lyn mNeonGreen.
- Source data 5. SP6 lyn-miRFP-pA.
- Source data 6. emilin3a GFP.
- Source data 7. emilin3a GFP-p2a-her6.
- Source data 8. emilin3a GFP-p2a-her9.
- Source data 9. emilin3a mScarlet-p2a-JICD.

### Data availability
Code is available under the MIT open source license on GitHub at: https://github.com/hsan-cheziranzo/notochord-lateral-inhibition (copy archived at swh:1:rev:2e5c5fe15e30ea6bacd-c0282e1506b44b05415af) (Sanchez-Iranzo, 2022). Images used for image analysis are available in Mendeley Data: https://doi.org/10.17632/fzmk5k982j.1 (CC BY 4.0).

The following dataset was generated:

| Author(s) | Year | Dataset title | Dataset URL | Database and Identifier |
|---|---|---|---|---|
| Sánchez-Iranzo H | 2022 | Strength of interactions in the Notch gene regulatory network determines patterning and fate in the notochord | https://doi.org/10.17632/fzmk5k982j.1 | Mendeley Data, 10.17632/fzmk5k982j.1 |

The following previously published dataset was used:

| Author(s) | Year | Dataset title | Dataset URL | Database and Identifier |
|---|---|---|---|---|
| Wagner DE, Weinreb C, Collins ZM, Megason SG, Klein AM | 2018 | Systematic mapping of cell state trajectories, cell lineage, and perturbations in the zebrafish embryo using single cell transcriptomics | https://www.ncbi.nlm.nih.gov/geo/query/acc.cgi?acc=GSE112294 | NCBI Gene Expression Omnibus, GSE112294 |

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

# Appendix 1

## Description of the theoretical model

### Lateral induction model

The lateral induction model was defined as a two-component system, Ligand (L) and Notch Intracellular Domain (NICD, represented as I in the equations). Notch-Ligand interaction in adjacent cells triggers the release of NICD following an increasing Hill function. NICD activates the expression of the ligand in its own cell following an increasing Hill function. The equations that describe the model are:

$$\frac{dI_i}{dt} = \beta_I \frac{\langle L_j \rangle^h}{a + \langle L_j \rangle^h} - \gamma_I \, I_i \tag{2}$$

$$\frac{dL_i}{dt} = \beta_L \frac{b I_i^h}{1 + b I_i^h} - \gamma_L \, L_i \tag{3}$$

$L_i$ and $I_i$ are the average concentrations of Ligand and NICD inside the cells, respectively. $\langle L_j \rangle$ is the average concentration of Ligand in each of the neighboring cells. $\beta_I$ and $\beta_L$ are the production rates of ligand and receptor, respectively. $\gamma_L$ and $\gamma_I$ are the degradation rates of Ligand and NICD, respectively, $a$ and $b$ the affinities, and $h$ is the Hill coefficient.

### Lateral inhibition model

This model is based on *Collier et al., 1996* and is similar to the lateral induction, with the only difference that the lateral inhibition model assumes that NICD activates the expression of a repressor that in turn inhibits the expression of the ligand. For this reason, the production of ligand is represented as an inhibitory Hill function.

The equations that describe the system are:

$$\frac{dI_i}{dt} = \beta_I \frac{\langle L_j \rangle^h}{a + \langle L_j \rangle^h} - \gamma_I \, I_i \tag{4}$$

$$\frac{dL_i}{dt} = \beta_L \frac{1}{1 + b I_i^h} - \gamma_L \, L_i \tag{5}$$

### Lateral inhibition model with mutual inhibition

The equations that describe this model are based on *Sprinzak et al., 2010*; *Sprinzak et al., 2011*

$$\frac{dN_i}{dt} = \beta_N - k_t N_i \langle L_j \rangle - k_c N_i L_i - \gamma N_i \tag{6}$$

$$\frac{dL_i}{dt} = \beta_L \frac{1}{1 + R_i^n} - k_t L_i \langle N_j \rangle - k_c N_i L_i - \gamma L_i \tag{7}$$

$$\frac{dR_i}{dt} = \beta_R \frac{(N_i \langle L_j \rangle)^n}{k_{RS} + (N_i \langle L_j \rangle)^n} - \gamma_R R_i \tag{8}$$

$N_i$, $L_i$ and $R_i$ are the average concentrations of Notch Receptor, Ligand and Repressor inside the cells, respectively. $\langle L_j \rangle$ and $\langle N_j \rangle$ are the average concentrations of ligand in the neighboring cells. $\beta_N$, $\beta_L$ and $\beta_R$ are the production rates of Notch Receptor, Ligand and Repressor, respectively. $\gamma$ and $\gamma_R$ are the degradation rates of Notch Receptor and Ligand/Repressor, respectively. $k_{RS}$ is the affinity, and $n$ is the Hill coefficient. $k_c$ and $k_t$ are the interaction strength between ligand and receptor in cis and *trans*, respectively. These two constants are referred as $K_{cis}$ and $K_{trans}$ in the manuscript.

### Simulations

All the visual simulations were generated by solving the equations using the Euler method with a step set to 0.01. Simulations were initialized with random values uniformly distributed between 0 and 0.1. To avoid boundary effects, we run simulations on a 100 cell array, where only the 20 central cells are displayed, while the 40 cells in each side buffer the boundary effect.

## Linear stability analysis

Linear stability analysis was done as previously described (**Sprinzak et al., 2011**). A prerequisite for pattern formation is the instability of the homogenous steady state ($N^*$, $L^*$, $R^*$), where every cell has the same value of $N_i$, $L_i$ and $R_i$. We first calculated the homogeneous steady state by making $N_i$ and $N_j$ equal to $N^*$, $L_i$ and $L_j$ equal to $L^*$, and $R_i$ equal to $R^*$, and solving the following system of equations (**Sprinzak et al., 2011**):

$$0 = \beta_N - k_t N^* L^* - k_c N^* L^* - \gamma N^* \tag{9}$$

$$0 = \beta_L \frac{1}{1 + (R^*)^n} - k_t L^* \langle N^* \rangle - k_c N^* L^* - \gamma L^* \tag{10}$$

$$0 = \beta_R \frac{(N^* \langle L^* \rangle)^n}{k_{RS} + (N^* \langle L^* \rangle)^n} - \gamma_R R^* \tag{11}$$

We solved these equations for the $R^*$, $L^*$ and $N^*$ using the fsolve function of the scipy.optimize python package.

The stability analysis requires the computation of the Jacobian matrix, that according to **Othmer and Scriven, 1971** can be expressed as $J = I_k \otimes H + M \otimes B$, where $I_k$ is the $k \times k$ identity matrix, $k$ is the number of cells, $\otimes$ represents the tensor product, $H_{ij} = \frac{\partial \dot{q}}{\partial q_j}$ is the change in production of species for a change in species $j$ in the same cell, $B_{ij} = \frac{\partial \dot{q}}{\partial \langle q_j \rangle}$ is the change in production of species for a change in species $j$ in a neighboring cell, and $M$ is the connectivity matrix defined as

$$M = \begin{cases} \frac{1}{2} \; if \; i \; and \; j \; are \; neighbours \\ 0 \; otherwise \end{cases}$$

In the specific case of our model, where cells are arranged unidimensionally, and $j$ are neighbors when $|i - j| = 1$.

The eigenvalues of $J$ Jacobian matrix are the eigenvalues of the various matrices $H + q_k BH$, where $q_k$ are the eigenvalues of the connectivity matrix $M$. For our particular $M$ matrix, $q_k$ values are always higher or equal to $-1$, meaning that we only need to compute an eigenvalue for the extreme case $q_k = -1$ to determine if the highest eigenvalue (known as the Maximum Lyapunov Exponent, MLE) has a positive real part.

Following this strategy, we computed the MLE value for a grid of $k_c$ and $k_t$ values logarithmically spaced between 0.001 and 100.

## Parameter values

| | |
|---|---|
| *Figure 1*, *Figure 1—figure supplement 1* | a = 0.1 <br> b = 10 <br> h = 2 <br> $\gamma_l$ = 1 <br> $\gamma_L$ = 1 <br> $\beta_l$ = 1 <br> $\beta_L$ = 1 |

*Continued on next page*

*Continued*

| | |
|---|---|
| *Figure 6*, *Figure 6—figure supplement 1* | $n = 1$<br>$k_c = 0.001$ to $100$<br>$k_t = 0.001$ to $100$<br>$\gamma = 1$<br>$\gamma_R = 1$<br>$k_{RS} = 1$<br>$\beta_N = 3$<br>$\beta_L = 10$<br>$\beta_R = 10$ |
| *Figure 6—figure supplement 2* | $n = 1$<br>$k_c = 0.001$ to $100$<br>$k_t = 0.001$ to $100$<br>$\gamma = 1$<br>$\gamma_R = 1$<br>$k_{RS} = 1$<br>$\beta_N = 3$<br>$\beta_L = 1.5$ to $10$<br>$\beta_R = 1.5$ to $10$ |

