## [Editor Report]

This manuscript presents computational and experimental results to study lateral inhibition patterning in the zebrafish notochord, identifying Jag1a as a crucial ligand and marker for vacuolated cell fate whereas her6 and her9 repress Jag1a. The results are complemented with numerical simulations of lateral induction and lateral inhibition circuits in one-dimensional arrays, together with linear stability analysis. The work is very well done and makes an important contribution to the understanding of notochord development.

---

## [Decision Letter]

[Editors' note: this paper was reviewed by Review Commons.]

---

## [Author Response]

Reviewer #1Evidence, reproducibility and clarity:SummaryIn their manuscript, the authors characterize the process of differentiation into vacuolated and sheath cells in the zebrafish Notochord. They convincingly show that this differentiation process is essentially governed by Notch mediated lateral inhibition in one dimension (along the AP axis of the notochord). The authors use imaging of fluorescent reporters, single cell sequencing, and genetic perturbations to identify the key components involved in the lateral inhibition feedback including the receptors, ligands, and the transcriptional repressors. They also use mathematical modeling to guide their experiments and show that strong cis-inhibition between receptors and ligands play an important role in the lateral inhibition process. This is one of the unique cases where lateral inhibition in 1D is identified and characterized. The establishment of the system for studying the process of lateral inhibition is important.Major commentsOverall, the manuscript is well written and describes a rigorous analysis of the lateral inhibition process in the system. The experiments performed follow a clear logic and conclusions drawn from the results are sound. Mathematical modeling is performed in a rigorous manner and the results are presented in a clear manner.

We thank the reviewer for the review of our manuscript and for stating that our data convincingly show our claims and that our work is important. Below we aim to address her/his concerns.

I do not think that additional experiments are required to support the claims presented, except a control for figure 3 (see below). However, some additional analysis of the data and some expansion of the modeling may provide additional support for the conclusions. These include:1. The alternating pattern of precursor cells in aim1 indeed seems matching lateral inhibition. However, it would be worth making some clarifications as well as analyzing some aspects expected from a lateral inhibition patterns:a. The pattern in Figure 1J,N show some dark gaps that are not green nor magenta. Are there cells in this gaps? If so, are these undecided cells? Would be good to clarify this point.

There are indeed cells in the dark gaps in Figure 1J, N. In the *in vivo* lineage tracing shown in Figure 1N, we can observe that most of these non-labelled cells become sheath cells. As tp1:GFP is not an endogenous gene, this suggests that the cells that are not labeled might not receive enough Notch signaling to activate the reporter. This highlights the importance of using endogenous markers. Indeed, we do not detect such gaps upon *her6*/*her9* + *jag1a* mRNA staining (Figure 2). To address this important point we have added supporting data and clarified this in the text (Section 2 results, lines 196-200).

b. While figure 1M shows a cross section of fluorescence signal, it does not reveal whether the fluorescence per cell. Are there cells that express both green and magenta? Would be good to present a graph showing the fluorescence per cell in both green and red, The ideal expectation is that most cells would be at either high Jag1/low GFP or vice versa.

We have added an additional plot in in Figure 1 —figure supplement 1G where we show the intensity per cell of a single plane of the image shown in Figure 1J. We observe in the quantification that there are no double positive cells. Moreover, we have added the quantification per pixel, that although noisier, shows a similar result (Figure 1 —figure supplement 1H).

c. In a 1D lateral inhibition pattern it is expected that each high jag1 cell would have only low Jag1 neighbors, but low jag1 cells can still be neighbors (as shown in the simulation). Would be good to show if this is indeed the case here.

Due to the high stability of the fluorescent proteins, quantifying *jag1a* intensity can lead to erroneous conclusions. Thus, we evaluated cell fate instead of jag1a intensity. Cell fate has the advantage that is binary: vacuolated (*jag1a*) or sheath (notch), eliminating possible ambiguities.

To that end, we took advantage of our image feedback microscopy set up, where we could follow individual cells back in time and thus link them to their future fate at the disc-shaped stage. We have never observed two consecutive disc-shape cells that become vacuolated cells. In contrast, cells that become sheath cells have adjacent cells that become sheath cells. We now show the results of this quantification in Results section 2 lines 208-211.

2. In Figure 2 the authors claim that her6/9 and jag1a mRNA levels exhibit anti correlations. The authors should perform quantitative analysis to show this.

We now quantitatively show that *her6*/*her9* mRNA anticorrelate with *jag1a* mRNA in Figures 2H, O.

3. In Figure 3-5 the authors use emillin3a driver to overexpress different genes. Is the expression of emillin3a uniform? It is hard to see from Figure S5C. Also, would be good to show that it is uncorrelated with Jag1a of tp1-GFP.

We now show higher magnification images of the notochord (Figures XX). Furthermore, we quantified how many tp1:GFP and *jag1a*:mNeonGreen are *emilin3a*:mScarlet-positive. Although not all the notochord cells have the same intensity of *emilin3a*:mScarlet, 98±3% of the tp1:GFP and 99±2% of the *jag1a*:mNeonGreen notochord cells express this marker, independently of these cells being sheath (tp1) or vacuolated (*jag1a*) cell precursors (Figure 3 —figure supplement 1 J–I).

4. In Figure 3h-J the authors show results with Crispr KO of Her6/9. It is important to show that the KO works and at what level, for example by showing mRNA levels after the KO.

To validate our experiments, we have performed PCR of the same embryos that we used for imaging. In Author response image 1, B we provide the agarose gel where is possible to see the high efficiency of gene deletion. Based on these data, we have included in our quantification only the embryos where both *her6* and *her9* knock out bands were detected. Moreover, we show below (Author response image 1, D) a comparison of our quantification using all the embryos or only those where both the her6 and her9 deletions have been confirmed by PCR. Additionally, we have sequenced the PCR products, confirming the deletion of those genes. Our results remain highly significant and we have updated Figure 3J and the methods section accordingly.

**Author response image 1. sa2fig1:** Validation of her6/her9 transient knock-out generation. (A, B), Confirmation by PCR of her6 (A) and her9 (B) gene deletion after the injection of the Cas9 protein together with guides targeting her6 and her9 genes. Expected PCR product after deletion: 519 bp for her6 and 225 bp for her9. In some cases, due to repair not exactly as expected, different PCR product size can be detected. Only embryos were a her6 and her9 bands were detected were included in the analysis. (C) Original plot including all the imaged fish. (D) New plot including only the samples labeled in green.

Also, a control showing gRNA for an unrelated gene is needed.

We now use a control guide that was specifically designed to not target any zebrafish gene, named as ‘universal guide’ (1) (Figure 3J).

In Figure 3J it is not clear what the y-axis means. Is this a number or a fraction? The authors should clarify this.

We thank the reviewer for pointing this out. We have now clarified in the legend of Figure 3 and Figure 1 —figure supplement 2 that we quantify the number of adjacent *jag1a*-positive cells. Because of the geometry of the tissue, the maximum value is 2.

5. Regarding the mathematical modeling showing that patterning occurs when Kcis>Ktrans. It would be important to show that this conclusion is valid when changing other parameters. In particular:a. Does this stay when cooperativity is assumes (n,m>1)

When we assume cooperativity, for example n, m = 2, patterning is also possible for Kcis < Ktrans. This agrees with published results (2–4). Importantly, we observe that with cooperativity, although a Kcis > Ktrans value is not required for patterning, it promotes patterning by making the homogeneous steady state more unstable. We have now added a new LSA analysis using different parameter values in a new supplementary figure (Figure 5 —figure supplement 2) and described our findings in detail in the last section of the results.

b. Does this conclusion stays when the relative expression of Notch and its ligand vary (if betaN>betaL or vice versa). I suspect that this is the case, but by showing that the conclusions from the model are valid for a larger parameter range would strengthen the claims.

We have also added new LSA plots with different relative values for betaN and betaL (Figure 5 —figure supplement 2), including betaN > betaL, betaN = betaL, and betaN < betaL. Interestingly, patterning is not always possible, but in all the cases, the MLE value is higher in a region where Kcis > Ktrans, indicating that a higher Kcis than Ktrans promotes patterning, independent of the value of other parameters.

6. Some minor comments/questions:a. Would be nice to check whether Her6/9 direct targets of Notch. While I don't think it's necessary to perform additional experiments to do that, it would be good to check if there binding sites for RBPJ in the regulatory regions of Her6/9.

A recent publication (5) has analyzed RBPJ target genes in the zebrafish tailbud, a region of the fish that includes part of the notochord. We now refer to this publication and show that there are RBPJ binding sites close to the her6/her9 transcription start sites (Figure 2 —figure supplement 1N).

b. Figure S7 is written as S5 by mistake

We appreciate that the reviewer noticed this typo. Moreover, we have now changed the formatting to follow *eLife* guidelines.

c. Figure 5A: color code mismatch (Red and blue arrows match the legend but not the titles circuit1 and 2)

We have corrected this mistake.

d. It seems there may be a confusion in the text between the reference to Figure 5C, D,F,G,I and to 5E,H,J

We have corrected this mistake.

The data and methods are sufficiently detailed and the statistics and statistical analysis are adequate.Significance: The main significance of the results in my mind are:1. Establishment of the notochord as a system for studying lateral inhibition. The authors did an excellent job identifying the components of the circuit and their regulatory relations.2. In particular, this is one of the relatively rare case showing lateral inhibition in an actual 1D configuration. I don't think there are many other characterized cases of lateral inhibition in 1D. The authors should point that out in the manuscript. I agree with the authors that this system would be a really nice system to quantitatively study lateral inhibition circuits in the future.3. The observation that stronger cis-inhibition is required for patterning is new and interesting.Overall the findings are new and the conclusions would certainly be interesting for the general community of developmental biology and in particular for researchers interested in quantitative developmental patterning processes (e.g. systems biology).This review is based on my expertise in analyzing Notch mediated patterning, both on the experimental and theoretical sides.Reviewer #2Evidence, reproducibility and clarity:The manuscript "Strength of interactions in the Notch gene regulatory network determines patterning and fate in the notochord" by H. Sanchez-Iranzo, A.Halavatvi and A. Diz-Muñoz presents computational and experimental results to study lateral inhibition patterning in the zebrafish notochord. The authors identify Jag1a as a crucial ligand and as a marker for vacuolated cell fate, in opposition to tp1 which the authors identify as a marker of sheath cell fate. The authors also identify her6 and her9 as repressors of jag1a, which are able to drive the sheath cell fate upon overexpression. They show that jag1a overexpression drives repression of her6/9 within the overexpressed cell and not in adjacent cells, suggesting jag1a is mediating cis-inhibition. These results are complemented with numerical simulations of lateral induction and lateral inhibition (with and without cis-inhibition) circuits in one dimensional arrays, together with linear stability analysis.My major criticism is that I do not agree with the conclusion that the results show that cis-interactions are required to be stronger than trans-interactions. The modeling results of Figure 5B correspond to no cooperativity, as the authors state, and indicate that cis-interactions need to be stronger than transinteractions. Indeed these results are in perfect agreement with previous ones from analogous and from similar models (Ref. 21 and 22; this agreement should be indicated). However, for higher cooperativities cis-interactions do not need to be stronger than trans anymore, as previously published computational results have also shown. Indeed the lateral inhibition circuit without cis-interactions is sufficient to drive patterning if cooperativity is high enough (Ref. 13). Therefore, unless the authors demonstrate that there is no cooperativity participating in the process of cell fate choice in the notochord, it can not be predicted that cis-interactions need to be greater than trans. Therefore, this part of the manuscript should be reformulated (e.g. move the modeling part after the experimental part, such that the modeling proposes that cooperativity is expected to be low). Regarding the experimental data, knockdown experiments of jag1a/b could reinforce the conclusion that cis interactions are stronger in the wt than trans interactions, albeit are not required in my opinion.

We thank the reviewer for the thorough review of our manuscript and for pointing out the importance of identifying a novel system to study lateral inhibition patterning. In the revision below we aim to clarify all caveats and better experimentally link our findings to the modelling framework we propose.

As the reviewer points out, if cooperativity is assumed, patterning is possible with no cis interactions. In our manuscript we show, for the first time, that in the absence of cooperativity, stronger cis than trans interactions are required for patterning. As discussed in R1Q5, we have performed an extensive LSA analysis where we study patterning with different parameter values, including cooperativity. We observe that, regardless of cooperativity, a Kcis > Ktrans value promotes patterning by making the homogeneous steady state more unstable. We have included this new data in the manuscript (Figure 5 —figure supplement 2), and reformulated our claims to ‘stronger cis than trans interactions promote patterning’. Moreover, as suggested by the reviewer, we have changed the order of the experimental and modeling part.

Other minor issues:a. In my opinion the role of jag1b should be further clarified. Despite its pattern of expression is more continuous than that of jag1a (as also indicated in Ref.28), results from Ref.28 showed that only when both jag1a and jag1b are knockdown, there are phenotypes of cell switch change. This may reflect that jag1b can take the role of jag1a in its absence or that- both are required in the wt for patterning. Hence, these hypotheses should be further investigated despite the more continuous pattern of expression. Alternatively, the manuscript should explicitly indicate that jag1b may be a relevant player that can not be completely discarded.

To study the role of these genes on patterning we have used previously validated *jag1a* and *jag1b* morpholinos (6). We have observed the strongest effect when both morpholinos are injected simultaneously (Figure 2D, E). In this experiment, tp1:GFP signal almost completely disappears specifically in the notochord, further supporting the fact that *jag1a* and *jag1b* are the most (if not the only) relevant Notch ligands in the notochord. In addition, we observe a spread of jag1a:mScarlet reporter to almost all the cells in the notochord (Figure 2D). This is probably the result of decreased Notch signaling, decreased *her6*/*her9* expression, and as a result no repression of *jag1a* regulatory regions. Altogether, these results show the importance of *jag1a*/*jag1b* in patterning the notochord.

When we injected *jag1a* or *jag1b* morpholinos individually, we observed a similar effect, but to a lower extent (Figure 2B, C and E). These results suggest that *jag1a* and *jag1b* have similar roles on notochord patterning, but neither can compensate the loss of the other. We now specifically state that in lines 174-179.

b. It is well known that lateral inhibition drives alternated cell fates, while lateral induction does not. Therefore, computational results like those of Figure 1F are very well known (e.g. very similar can be found in Ref. 13). Those of Figure 1E are also known (Boareto et al. J R Soc Interface 2016; Matsuda et al. Science Signaling (2012), Ref. 49). Hence, the computational results from Figure 1E and F seem unnecessary to be included. The authors could just state the conclusions by referencing the literature. Yet, if they are preserved for clarity, references to the literature and emphasis that it is already known what each of these circuits drive has to be included.

We think that Figure 1 E and F are helpful to follow the results. Thus, we have rephrased this paragraph to make clearer that lateral inhibition and lateral induction networks have already been described in previous publications.

c. The authors analyse single cell RNA-seq data. How is the expression of jag1a, jag1b and tp1 in the cells where her6 and her9 are expressed and in those where emilin3a is expressed but not her6 nor her9?

The single-cell RNA-Seq data that we used is very helpful to identify genes that are expressed in the notochord when we group the cells. However, the depth of sequencing is too low to allow us to make reliable conclusions about specific cells. To address this reviewer’s comment, we have included new panels where we show *her6*, *her9*, *jag1a*, *jag1b* and *emilin3a* expression in each of the cells (Figure 2 —figure supplement 1A – F). In these plots, we can observe that emilin3a can be detected in most of the cells selected as notochord. However, it is difficult to make conclusions about the rest of the genes that are expressed at lower levels.

On the other hand, it is worth noting that tp1 is not an endogenous gene, but a regulatory sequence that has been shown to be responsive to Notch signaling. Thus, its expression cannot be evaluated in the single-cell RNA-Seq. We now further clarify this in the text (Section 1 results lines 162-163 and Methods lines 512-515).

d. Figure 1N: at which hpf?

We have added the stage in the figure legend (24 hpf).

e. Figure 3A-F: why when her6 is overexpressed there seems to be in total more cells expressing jag1? (Scarlet colour is much more present in images D and F than in B)

We thank the reviewer for pointing this out. Indeed, the image shown in Figure 3B was not very representative. We have replaced it by a more representative image.

f. Figure 5C-H: in embryos where jag1 is overexpressed, there seems to be less overall expression of her6 and her9 compared to control (images G and H show much less red and cyan intensity than D and E). How the authors explain this?

We thank the reviewer for pointing out this artefact. This was caused by the non-optimal adjustment of channel intensities. We have now adjusted better the intensities. Nevertheless, it is worth noting that the images used for quantification were adjusted automatically within the python pipeline already in the original version of the manuscript. Specifically, the intensities of each of the channels were normalized between 0 and 1, where 0 was assigned to the minimum intensity value in the image, and 1 to the maximum value.

g. The first time tp1 appears mentioned in the manuscript, it should be introduced: i.e. to indicate what it is and what it is known about it.

We have now added a sentence introducing the tp1 promoter (Section 1 results lines 162-163 and Methods lines 512-515).

h. In section 1 of results, where it indicates "We first modelled a lateral inhibition network as " should be changed to "We first modelled a lateral induction network as ".

We thank the reviewer for pointing out this typo. We have now corrected it in the manuscript.

Significance:My expertise is on modeling. The experimental results of the manuscript are novel to my knowledge and provide a rather complete framework to characterize and understand binary cell fate choices (vacuolated and sheath cell fates) in the notochord of zebrafish embryos. As the authors indicate, ref. 28 (from 2010), established that Notch signaling by Jag1 (1a and 1b), and through her9, control the cell fate switch between vacuolated and sheath cell fates. In this manuscript, the authors add another HER/HES actor, her6, and show its complementary pattern with jag1a and that it can repress it. In addition, the authors conceptualize their finding within the framework of lateral inhibition and propose jag1a as the relevant player. The manuscript is very well organized and written. The results seem appropriately analyzed and the modeling framework and results seem correct. Yet, the computational results are less novel (see report above). Overall I find it a clear manuscript that clarifies and sets a new system to study lateral inhibition patterning. It can be very well suited to a developmental biology audience.Reviewer #3Evidence, reproducibility and clarity:Summary:The paper proposes a new model for the specification of vacuolated vs. sheath cell fate in the zebrafish notochord. In doing so, it further establishes a model system for understanding the molecular mechanisms underpinning cell fate decision making in vertebrate development, and to explore the regulatory relationships of the Notch signalling pathway. In particular, the study reveals how Jag1 can not only function in the context of a lateral induction mechanism, but also in producing lateraly inhibition when inhibited by her6 and her9. The authors also clone an emilin3a promotor element that is sufficient to drive robust expression in the notochord of zebrafish embryos.

We thank the reviewer for the review of our manuscript and for highlighting the importance of establishing the notochord as a model system to better explore the mechanism of action of Notch signaling in a vertebrate context. Below we aim to address her/his concerns.

Major comments:– Are the key conclusions convincing? The conclusions are convincing and well supported by experimental data.

– Should the authors qualify some of their claims as preliminary or speculative, or remove them altogether? All claims are valid and well supported, assuming suitable n numbers are presented for the functional experiments in Figures 3 and 4.

– Would additional experiments be essential to support the claims of the paper? Request additional experiments only where necessary for the paper as it is, and do not ask authors to open new lines of experimentation. The existing conclusions of the paper would be significantly strengthened by loss-offunction experiments for Jag1.

As described in the answer to point “a” from reviewer 2:

“To study the role of these genes on patterning we have used previously validated *jag1a* and *jag1b* morpholinos (6). We have observed the strongest effect when both morpholinos are injected simultaneously (Figure 2D, E). In this experiment, tp1:GFP signal almost completely disappears specifically in the notochord, further supporting the fact that *jag1a* and *jag1b* are the most (if not the only) relevant Notch ligands in the notochord. In addition, we observe a spread of jag1a:mScarlet reporter to almost all the cells in the notochord (Figure 2D). This is probably the result of decreased Notch signaling, decreased *her6*/*her9* expression, and as a result no repression of *jag1a* regulatory regions. Altogether, these results show the importance of *jag1a*/*jag1b* in patterning the notochord.

When we injected *jag1a* or *jag1b* morpholinos individually, we observed a similar effect, but to a lower extent (Figure 2B, C and E). These results suggest that *jag1a* and *jag1b* have similar roles on notochord patterning, but neither can compensate the loss of the other. We now specifically state that in lines 174-179.”

– Are the suggested experiments realistic in terms of time and resources? It would help if you could add an estimated cost and time investment for substantial experiments. Approximately 3 months would be required to generate morpholinos against Jag1, and to perform experiments alongside appropriate controls to demonstrate phenotypic rescue.

– Are the data and the methods presented in such a way that they can be reproduced? Yes, the results are very well presented with an adequate description of the methods.

– Are the experiments adequately replicated and statistical analysis adequate? This is not clear for Figures 3 and 4. The n numbers need to be clearly written in the figure legends, as for the previous figures.

We have now explicitly mentioned the numbers of fish for Figures 3 and 4 in the figure legends, as well as their corresponding supplementary figures.

Minor comments:

– Specific experimental issues that are easily addressable. N/A

– Are prior studies referenced appropriately? Yes

– Are the text and figures clear and accurate? Yes.

– Do you have suggestions that would help the authors improve the presentation of their data and conclusions?1. Results section 4: It would be helpful to describe more details about her6 and her 9 gene deletions.

We describe in more detail the design of her6 and her9 deletions in a new panel in Figure 2 —figure supplement 1N.

2. Results section 5, paragraph 1. The primary evidence is a loss of vaculated cell fate, with an assumed increase in sheath cells. The conclusion should be re-written accordingly.

We thank the reviewer for pointing this out and agree that our conclusions were misleading. We have clarified this in the text and in the figure legend (Figure 4) by defining the proportion of vacuolated cells as the number of vacuolated cells divided by the sum of sheath and vacuolated cells. Thus, a lower proportion of vacuolated cells is equivalent to a higher proportion of sheath cells.

3. Results section 6, paragraph 1. The overexpression of full length ligand assumes that Notch levels are not limiting in the system to be able to observe and effect. Have the authors considered this alternative explanation for an absence in phenotype?

We indeed cannot exclude the possibility that Notch levels are limiting. We now discuss this possibility in the text (Section 6 results, lines 326-327). However, it is worth noting that both the results of our modelling and prior knowledge from the mammalian and *Drosophila* homologs (Jag1/Serrate have been shown to be weak Notch activators, but strong cis-inhibitors) support our conclusion that stronger cis than trans interactions is a relevant mechanism of action of Notch signaling in vertebrates.

Significance– *Describe the nature and significance of the advance (e.g. conceptual, technical, clinical) for the field:* The Notch pathway is an essential signalling pathway in multiple contexts and it is therefore very important to increase our understanding of its mechanism of action. This work presents interesting findings in this regard, and further establishes a model system to better explore its mechanism of action in a vertebrate context.

– *Place the work in the context of the existing literature (provide references, where appropriate):* Notch function has been studied in the context of notochord development here:Norman, J., Sorrell, E.L., Hu, Y., Siripurapu, V., Garcia, J., Bagwell, J., Charbonneau, P., Lubkin, S.R., and Bagnat, M. (2018). Tissue self-organization underlies morphogenesis of the notochord. Philos. Trans. R. Soc. B Biol. Sci. 373.Yamamoto M, Morita R, Mizoguchi T, Matsuo H, Isoda M, Ishitani T, Chitnis AB, Matsumoto K, Crump JG, Hozumi K, Yonemura S, Kawakami K, Itoh M. Mib-Jag1-Notch signalling regulates patterning and structural roles of the notochord by controlling cell-fate decisions. Development. 2010 Aug 1;137(15):2527-37. doi: 10.1242/dev.051011. Epub 2010 Jun 23. PMID: 20573700; PMCID: PMC2927672.

– *State what audience might be interested in and influenced by the reported findings:* Zebrafish developmental biologists, anyone interested in Notch signalling.

– *Define your field of expertise with a few keywords to help the authors contextualize your point of view. Indicate if there are any parts of the paper that you do not have sufficient expertise to evaluate:* Zebrafish developmental biology and notochord development.

References

1. W. A. Wierson, *et al.*, Efficient targeted integration directed by short homology in zebrafish and mammalian cells. *ELife* 9, 1–25 (2020).

2. D. Sprinzak, *et al.*, Cis-interactions between Notch and Δ generate mutually exclusive signalling states. *Nature* 465, 86–90 (2010).

3. D. Sprinzak, A. Lakhanpal, L. LeBon, J. Garcia-Ojalvo, M. B. Elowitz, Mutual inactivation of Notch receptors and ligands facilitates developmental patterning. *PLoS Comput. Biol.* 7 (2011).

4. P. Formosa-Jordan, M. Ibañes, Competition in notch signaling with cis enriches cell fate decisions. *PLoS One* 9 (2014).

5. Z. Ye, C. R. Braden, A. Wills, D. Kimelman, Identification of in vivo Hox13-binding sites reveals an essential locus controlling zebrafish brachyury expression. *Dev.* 148 (2021).

6. M. Yamamoto, *et al.*, Mib-Jag1-Notch signalling regulates patterning and structural roles of the notochord by controlling cell-fate decisions. *Development* 137, 2527–2537 (2010).